# Differential ion dehydration energetics explains selectivity in the non-canonical lysosomal K⁺ channel TMEM175

**SeCheol Oh[1†], Fabrizio Marinelli[2†], Wenchang Zhou[2†], Jooyeon Lee[3], Ho Jeong Choi[3], Min Kim[3], José D Faraldo-Gómez[2*], Richard K Hite[1*]**

[1]Structural Biology Program, Memorial Sloan Kettering Cancer Center, New York, United States; [2]Theoretical Molecular Biophysics Section, National Heart, Lung and Blood Institute, National Institutes of Health, Bethesda, United States; [3]Department of Chemistry, Chungbuk National University, Cheongju-si, Republic of Korea

**Abstract** Structures of the human lysosomal K⁺ channel transmembrane protein 175 (TMEM175) in open and closed states revealed a novel architecture lacking the canonical K⁺ selectivity filter motif present in previously known K⁺ channel structures. A hydrophobic constriction composed of four isoleucine residues was resolved in the pore and proposed to serve as the gate in the closed state, and to confer ion selectivity in the open state. Here, we achieve higher-resolution structures of the open and closed states and employ molecular dynamics simulations to analyze the conducting properties of the putative open state, demonstrating that it is permeable to K⁺ and, to a lesser degree, also Na⁺. Both cations must dehydrate significantly to penetrate the narrow hydrophobic constriction, but ion flow is assisted by a favorable electrostatic field generated by the protein that spans the length of the pore. The balance of these opposing energetic factors explains why permeation is feasible, and why TMEM175 is selective for K⁺ over Na⁺, despite the absence of the canonical selectivity filter. Accordingly, mutagenesis experiments reveal an exquisite sensitivity of the channel to perturbations that mitigate the constriction. Together, these data reveal a novel mechanism for selective permeation of ions by TMEM175 that is unlike that of other K⁺ channels.

**\*For correspondence:**
jose.faraldo@nih.gov (JDF-G);
hiter@mskcc.org (RKH)

[†]These authors contributed equally to this work

## Editor's evaluation

This manuscript explores the mechanisms of permeation and selectivity in the unusual potassium-selective ion channel TMEM175, which lacks a canonical selectivity filter. The study is led by molecular dynamics simulations and free energy calculations, complemented by a cryoEM analysis and electrophysiological recordings. The authors propose a novel, single ion-based mechanism of permeation, together with a partial dehydration-driven selectivity mechanism. This study will appeal to readers interested in the structure and function of ion channels and in molecular mechanisms of ion translocation.

## Introduction

Transmembrane protein 175 (TMEM175) is a potassium (K⁺)-selective cation channel that is evolutionarily distinct from all other known ion channels. In mammals, TMEM175 is expressed in lysosomes, where it is critical for maintaining lysosomal homeostasis (*Cang et al., 2015*). As a K⁺-selective channel, TMEM175 contributes to establishing the membrane potential across the lysosomal membrane (*Cang et al., 2015*). Loss of TMEM175 function can lead to numerous defects in the endolysosomal system including dysregulation of lysosomal pH, defects in autophagy and mitophagy, reduced

glucocerebrosidase activity, and an increased susceptibility to α-synuclein cytotoxicity (*Jinn et al., 2019*; *Jinn et al., 2017*; *Krohn et al., 2020*). Dysregulation of lysosomal function is commonly associated with neurodegenerative disease and point mutations in TMEM175 have been identified that are strongly associated with development of Parkinson's disease (PD) (*Blauwendraat et al., 2019*; *Iwaki et al., 2019*; *Jinn et al., 2019*; *Jinn et al., 2017*; *Krohn et al., 2020*; *Nalls et al., 2014*; *Wie et al., 2021*). For example, the M393T point mutation is a loss-of-function mutation associated with the increased likelihood and early onset of PD and the Q65P mutant is a gain-of-function mutation associated with a reduced likelihood of developing of PD (*Wie et al., 2021*).

Consistent with being evolutionarily distinct from other ion channels, TMEM175 channels from prokaryotes and humans adopt a unique architecture relative to all known ion channel structures (*Brunner et al., 2020*; *Lee et al., 2017*; *Oh et al., 2020*). Structures of two prokaryotic TMEM175 homologs in closed states revealed a homotetrameric channel possessing a central ion conduction pathway lined by several layers of hydrophobic residues from the first transmembrane (TM) helix of each of the four protomers (*Brunner et al., 2020*; *Lee et al., 2017*). Despite similar overall structures, analysis of the two prokaryotic TMEM175 structures led to conflicting proposals for how TMEM175 channels selectively permeate $K^+$ ions (*Brunner et al., 2020*; *Lee et al., 2017*). However, human TMEM175 (hTMEM175) channels are homodimers, rather than homotetramers, and are ~30-fold selective for $K^+$ over $Na^+$, whereas prokaryotic TMEM175 channels exhibit only 2- to 4-fold selectivity for $K^+/Na^+$ (*Cang et al., 2015*). Because of the notable structural and functional differences between human and prokaryotic TMEM175, it remains unclear if the mechanisms gleaned from analyses of prokaryotic TMEM175 channels are generalizable for hTMEM175.

To begin to understand the mechanisms that underlie TMEM175 function in lysosomes, we determined the structure of hTMEM175 by cryo-EM and identified two conformational states (*Oh et al., 2020*). hTMEM175 protomers consist of two homologous repeats of six TM helices, which assemble into a dimer to form the channel. A 2-fold symmetric ion conduction pathway runs along the center of the dimer, lined by two copies of TM1 and TM7, the first helices of each repeat. Unlike most cation channels, including canonical $K^+$ channels such as KcsA and Shaker, the pore of TMEM175 contains no clusters of hydrophilic side chains or backbone atoms that control selectivity (*Doyle et al., 1998*; *Long et al., 2005*). Rather, the most striking feature in the pore of TMEM175 is a narrow hydrophobic constriction formed by the side chains of the conserved Ile46 from TM1 and the conserved Ile271 from TM7, which we termed the isoleucine constriction (*Oh et al., 2020*). In one of the two cryo-EM structures, the isoleucine constriction is too narrow to accommodate $K^+$ ions and thus we assigned it as a closed state. In the other structure, however, the isoleucine constriction is sufficiently dilated to accommodate partially hydrated $K^+$ ions, and therefore we proposed that this structure corresponds to a conductive state of the channel. Moreover, because ions would need to be partially dehydrated to traverse the pore, it was proposed that the relative energetics of $K^+$ and $Na^+$ dehydration through the isoleucine constriction contributes to the ion selectivity of TMEM175 (*Lee et al., 2017*; *Oh et al., 2020*). While plausible, this proposal relies on the assumption that the putative open state is indeed conductive; however, assigning functional states to cryo-EM structural snapshots is challenging. Here, we employ high-resolution cryo-EM structure determination, molecular dynamics (MD) simulations, and electrophysiological analyses to investigate the mechanisms of ion permeation and selectivity in TMEM175.

## Results

### Improved cryo-EM structures of hTMEM175

Taking advantage of recent advances in cryo-EM image analysis, we reprocessed our published cryo-EM images of TMEM175 in KCl to achieve higher resolution (*Punjani and Fleet, 2021*; *Punjani et al., 2020*; *Zivanov et al., 2018*). The resolution of the putative open state improved from 2.64 to 2.45 Å and the closed state improved from 3.03 to 2.61 Å (*Figure 1—figure supplement 1* and *Appendix 1—table 1*). While the improved structures are similar to those previously determined with an all-atom RMSD of 0.3 Å for the open state and 0.4 Å for the closed state, the higher-resolution reconstructions allowed us to better define the ion conduction pathways (*Figure 1A–D*). In both states, the ion conduction pathway is 2-fold symmetric, narrow, and largely devoid of polar contacts, particularly near the center due to the presence of several hydrophobic side chains including Ile46,

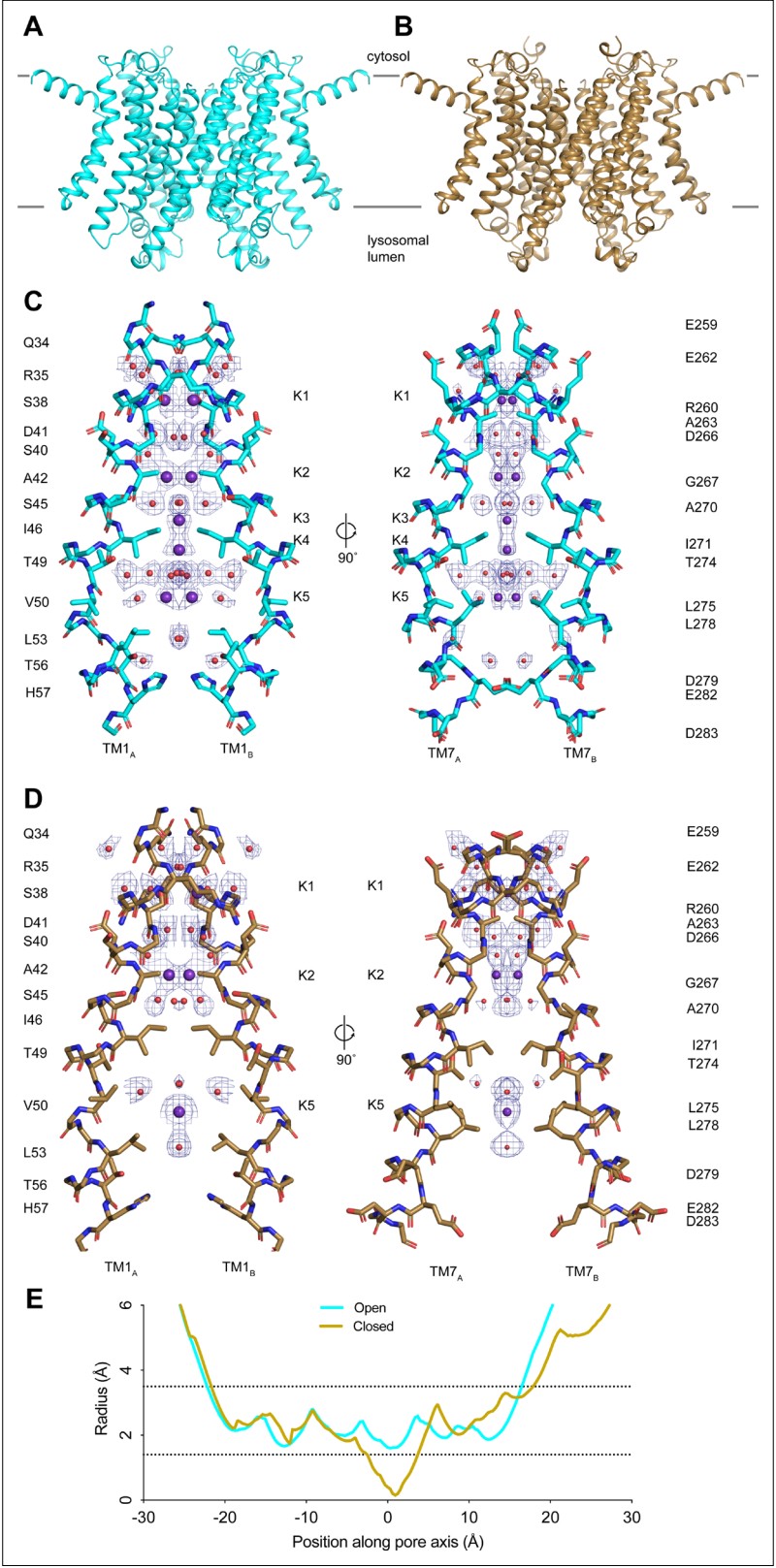

**Figure 1.** Structures of transmembrane protein 175 (TMEM175) in open and closed states. (**A–B**) Structures of the human TMEM175 in open (**A**) and closed (**B**) states. The channel is orientated with the cytosolic side facing up and the luminal side down. Gray bars represent approximate width of the membrane. (**C–D**) Ion conduction pathways in open (**C**) and closed (**D**) states. Pore-lining residues are shown as sticks. Bound K⁺ ions and water molecules are

*Figure 1 continued on next page*

*Figure 1 continued*

shown as purple and red spheres, respectively. Density peaks corresponding to K$^+$ and water molecules are shown as blue mesh and contoured at 12$\sigma$. Front and rear domains are removed for clarity. (**E**) Plot of pore radius as a function of position along the pore axis. The 0 position corresponds to the isoleucine constriction, positive values correspond to the luminal side of the pore, and negative values correspond to the cytosolic side of the pore.

The online version of this article includes the following figure supplement(s) for figure 1:

**Figure supplement 1.** Structure determination of human transmembrane protein 175 (TMEM175) in open and closed states.

**Figure supplement 2.** Structures of the isoleucine constriction sites in open and closed transmembrane protein 175 (TMEM175).

---

Val50, Leu53, Ile271, Leu275, and Leu278 (*Figure 1C–D*). In the closed state, the isoleucine constriction formed by Ile46 and Ile271 is too narrow to allow ion permeation (*Figure 1D*). The side chains of Ile271 point toward the channel axis, narrowing the pore to a minimum radius of 0.2 Å (*Figure 1D and E*). In the open state, by contrast, conformational changes in the pore-lining helices TM1 and TM7 helices dilate the isoleucine constriction to a minimum radius of 1.6 Å, which appears to be sufficient to accommodate a partially dehydrated K$^+$ ion (*Figure 1C and E*). In its dilated conformation, the isoleucine constriction adopts a pseudo 4-fold symmetric configuration with the side chains of both Ile46 and Ile271 adopting the *mt* rotamer (*Figure 1—figure supplement 2*).

The higher-resolution reconstructions also allowed an improved modeling of five ion-binding sites, which we call K1 through K5, and ordered water molecules in the pore of the open state. Two non-protein densities are now resolved at the K3 and K4 sites that flank the isoleucine constriction (*Figure 1—figure supplement 2*). We assigned these densities as K$^+$ ions based on our previous comparison with a structure determined in the presence of Cs$^+$ (*Oh et al., 2020*). The K3 and K4 ion-binding sites are positioned 3.7–4.0 Å away from the side chains of Ile46 and Ile271 and are coordinated exclusively by water molecules (*Figure 1—figure supplement 2*). Notably, the ions in the K3 and K4 sites are only partially hydrated. The K3-binding site on the cytoplasmic site of the constriction is coordinated by four ordered water molecules that are between 2.8 and 3.0 Å away, while the K4 site on the luminal side is coordinated by four water molecules that are between 2.9 and 3.5 Å away. The distance between the two sites is 2.8 Å and thus they are unlikely to be simultaneously occupied by ions.

For the closed state, we can now model three ion-binding sites and 25 water molecules into non-protein densities (*Figure 1D*). In the cytosolic region of the pore, where the channel undergoes minimal conformational changes during gating, the K1 and K2 ion-binding sites and water molecules occupy almost identical positions to those resolved in the open state (*Figure 1—figure supplement 2*). In contrast, there is little correspondence between the ion and water configurations at the isoleucine constriction or on the luminal side of the pore due to the conformational changes that occur to the channel during channel gating. The inward movement of Ile271 displaces the ion-binding sites on either side of the isoleucine constriction. Movements of Thr49 and Thr274, which both coordinate water molecules in the open state, result in a movement of the luminal K5 ion-binding site toward the luminal entrance of the pore. Thus, the higher-resolution structures reveal how changes in the protein structure alter the ion conduction pathway to prevent ion permeation in the closed state.

## Energetics and mechanism of K$^+$ permeation

To ascertain whether the structure of the putative open state is indeed permeable to K$^+$, we turned to all-atom MD simulations of the channel in a model phospholipid bilayer (*Figure 2—figure supplement 1*). Specifically, using the enhanced-sampling methodology known as multiple-walker Metadynamics (*Raiteri et al., 2006*), we induced the permeation of K$^+$ across the channel at 0 mV and symmetric 100 mM KCl and evaluated the associated potential of mean force (or free-energy profile). Importantly, these calculations used a newly formulated reaction coordinate (Materials and methods) designed to avoid a priori assumptions about the mechanism of permeation, such as the number of ions that reside within the pore at a given time.

The results from this analysis are summarized in *Figures 2 and 3*. We observe that K$^+$ ions reach the isoleucine constriction readily from either side of the membrane, through a series of shallow free-energy barriers and transient-binding sites (*Figure 2A*). At the isoleucine constriction, however, the

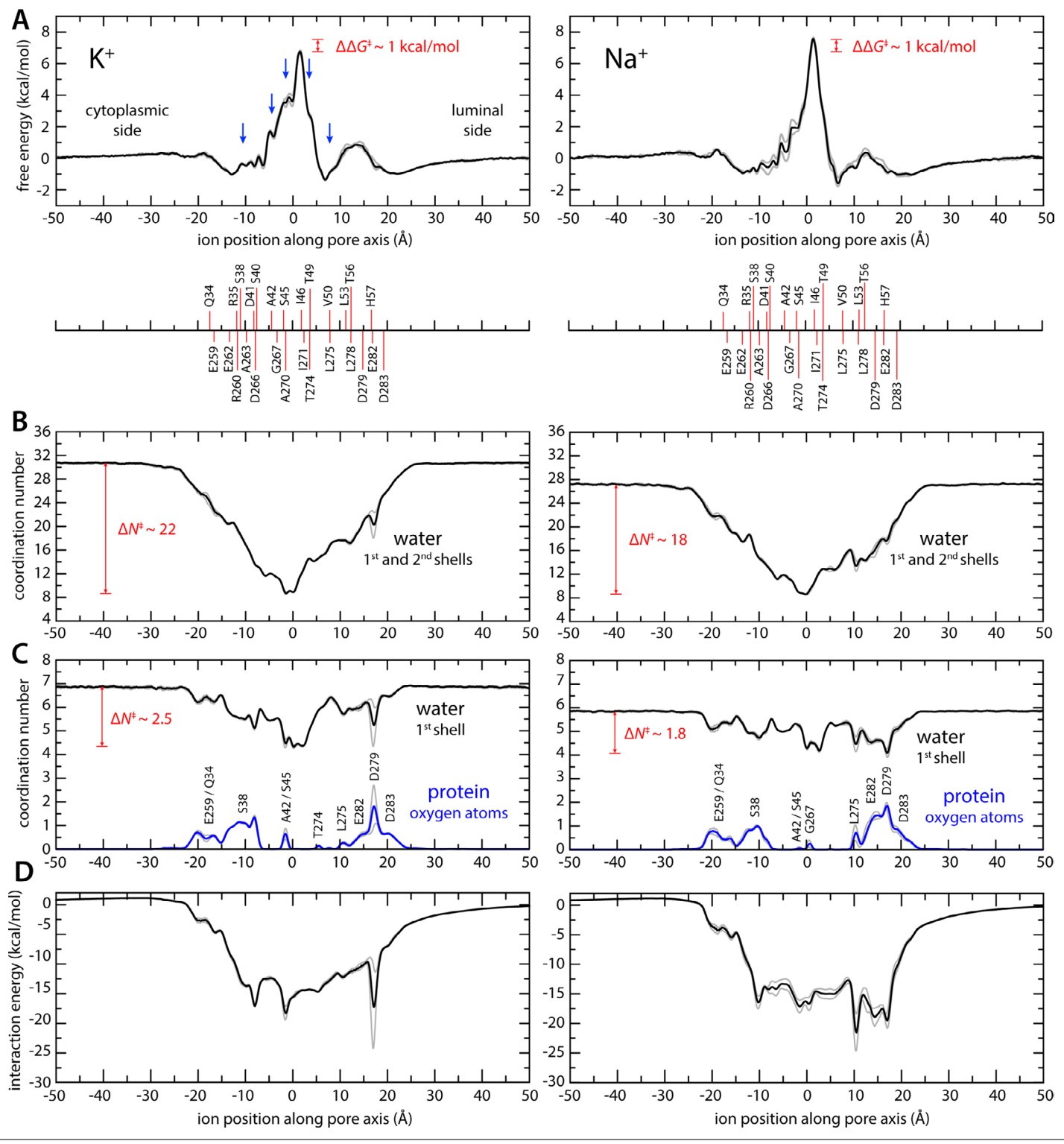

**Figure 2.** Enhanced-sampling molecular dynamics (MD) simulations of K+ and Na+ permeation through transmembrane protein 175 (TMEM175). (**A**) Potential of mean force (or free energy) as a function of the ion position along the pore axis (*black*), for either K+ (*left*) or Na+ (*right*). See also *Figure 2—figure supplement 1*. The difference between the free-energy peak for Na+ and that for K+ ($\Delta\Delta G$) is indicated. Blue arrows indicate the approximate position of the density peaks assigned to K+ ions in *Figure 1*. For reference, the position of selected residues in helices TM1 and TM7 is also indicated under the free-energy curves. These positions are defined as the time average of the center-of-mass of side chain and Cα atoms, including equivalent protein subunits, projected along the pore axis. (**B**) Average number of water molecules in the first and second ion hydration shells,

*Figure 2 continued on next page*

*Figure 2 continued*

as a function of the ion position along the pore axis (*black*). The degree of depletion at the isoleucine constriction (Δ*N*), relative to bulk hydration, is indicated for each ion. (**C**) Same as (**B**), only for the first ion hydration shell (black). The average number of protein-oxygen atoms observed to coordinate an ion, as a function the ion position, is also shown alongside (blue), indicating the principal contributing residues. See also *Figure 2—figure supplement 3*. (**D**) Ion-protein electrostatic interaction energy as a function of the ion position along the pore axis. In all panels, gray profiles represent the same quantity shown in black/blue calculated using only the first or second half of the simulation data. Comparison of these two profiles provides a metric of the statistical error in these calculations. The mean difference between the free-energy profiles for K$^+$ is 0.14 kcal/mol; for Na$^+$, it is 0.22 kcal/mol. See Materials and methods for further details and definitions.

The online version of this article includes the following figure supplement(s) for figure 2:

**Figure supplement 1.** All-atom molecular dynamics (MD) simulations of human transmembrane protein 175 (hTMEM175).

**Figure supplement 2.** Enhanced-sampling molecular dynamics (MD) simulations of K$^+$ and Na$^+$ permeation through human transmembrane protein 175 (hTMEM175).

**Figure supplement 3.** Enhanced-sampling molecular dynamics (MD) simulations of K$^+$ and Na$^+$ permeation through human transmembrane protein 175 (hTMEM175).

free energy increases steeply, peaking at about 7 kcal/mol (*Figure 2A*). These features recapitulate the pattern of non-protein densities resolved in the pore in the reconstruction of the putative open state; densities near Ser38 and Ala42 on the cytoplasmic side and near Val50 on the luminal side appear as metastable states in the calculated free-energy profile. Densities right below and above the central isoleucine constriction also appear as shoulders in the free-energy profile. Unlike in canonical K$^+$ channels, however, the simulation data reveals no evidence of a multi-ion process involving concurrent occupancy of proximal-binding sites near the constriction (*Figure 2—figure supplement 2*). Instead, K$^+$ ions approach and cross this constriction individually. It is apparent, too, that permeation requires a gradual but drastic depletion of the ion hydration shells. As K$^+$ traverses the isoleucine constriction, the first two hydration shells are reduced to only ~9 molecules, down from 31 in the bulk; the first shell averages to about 4, down from 7 (*Figures 2B, C , and 3A*).

Given the largely hydrophobic nature of the TMEM175 pore, it is logical that the free-energy barrier for permeation is significantly higher than those in canonical K$^+$ channels. The finding that this barrier is only 7 kcal/mol is however intriguing, given that the cost of K$^+$ dehydration is much larger than that (up to about 80 kcal/mol for full dehydration). This observation indicates that other factors must facilitate ion permeation in TMEM175. Indeed, further examination of the simulated trajectories using Poisson theory (Materials and methods) reveals that acidic side chains at both the luminal and cytosolic entrances of the pore generate a strongly favorable electrostatic field that spans the length of the pore (*Figure 2D*). Some of these residues also transiently coordinate the permeating ions as they enter and exit the pore, along with a few other carbonyl and carboxyl groups in side chains and backbone (*Figure 2C*, *Figure 2—figure supplement 3*). However, for a span of about 15 Å centered at the isoleucine constriction, direct protein contacts are minimal. That the barrier for ion conduction in TMEM175 is surmountable is thus a consequence of the degree of hydration retained by the permeating ions as well as the electrostatic field generated by the protein that attracts them to the pore interior.

Integration of the potential-of-mean-force profile obtained at 0 mV and 100 mM KCl translates into a single-channel conductance of approximately 0.23 pS (Materials and methods). At 500 mV, and assuming the structure of the channel is completely unresponsive to voltage, this conductance would in turn produce a 0.1 pA current, or a rate of K$^+$ permeation of about 0.6 ions/μs. To assess these estimates, we calculated an independent 1-μs-long MD simulation using conventional sampling but under 500 mV (lumen positive) and 400 mM KCl. This trajectory revealed a stable pore conformation and two K$^+$ permeation events, confirming that the open state is functionally conductive and validating the single-ion mechanism inferred from the free-energy simulations (*Figure 3C and D*; *Figure 3—video 1* and *Figure 3—video 2*). K$^+$ ions are seen to dwell at the shallow-binding site adjacent to the constriction, on the luminal side, before returning to the bulk or rapidly traversing the free-energy barrier, along with 4 water molecules in the first hydration shell (*Figure 3C*).

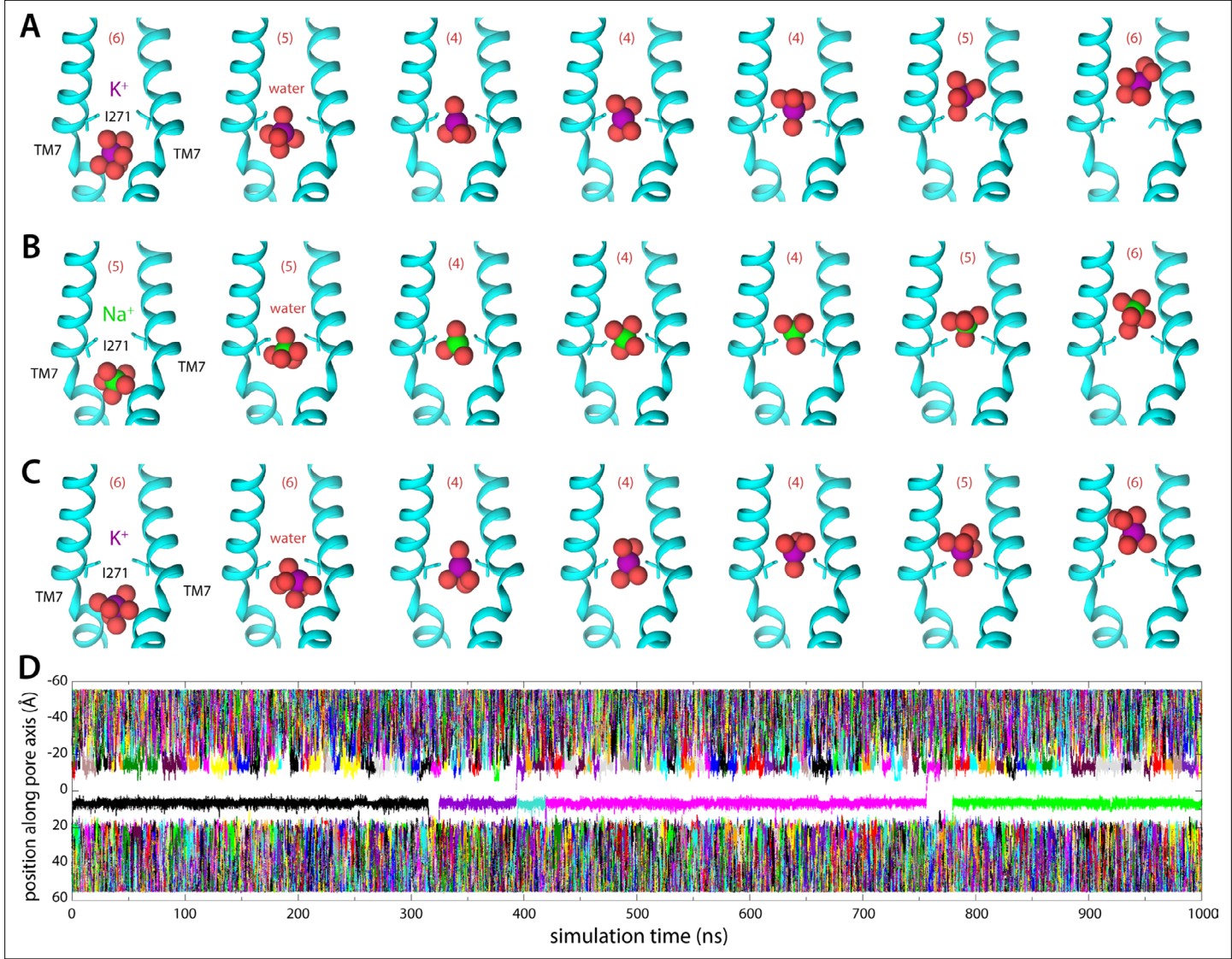

**Figure 3.** Molecular dynamics (MD) simulations of K$^+$ and Na$^+$ permeation through human transmembrane protein 175 (hTMEM175). (**A**) Snapshots of one of the trajectories calculated with enhanced-sampling MD, at 0 mV and 100 mM KCl, showing a K$^+$ permeation event across the isoleucine constriction. For clarity only TM7 is shown, in cartoon representation, alongside the side chain of I271. Red spheres represent the oxygen atoms of the water molecules in the first ion hydration shell, defined as in *Figure 2*. The number of water molecules in the first ion hydration shell is shown in parentheses. (**B**) Same as (**A**), for a Na$^+$ permeation event. (**C**) Same as (**A**), from a trajectory calculated with conventional MD sampling, but at 500 mV and 400 mM KCl. (**D**) For the latter trajectory, time series of the position along the channel axis of the subset of the K$^+$ ions in the simulation found within 10 Å of that axis. The trajectory revealed two permeation events in 1 µs.

The online version of this article includes the following video for figure 3:

**Figure 3—video 1.** Molecular dynamics (MD) simulation of K$^+$ permeation through human transmembrane protein 175 (hTMEM175) under an applied voltage.

https://elifesciences.org/articles/75122/figures#fig3video1

**Figure 3—video 2.** Molecular dynamics (MD) simulation of K$^+$ permeation through human transmembrane protein 175 (hTMEM175) under an applied voltage.

https://elifesciences.org/articles/75122/figures#fig3video2

## Differentials in ion dehydration relative to bulk explain selectivity for K$^+$ over Na$^+$

Like canonical K$^+$ channels, TMEM175 channels are more permeable to K$^+$ than Na$^+$. However, in the absence of the selectivity filter seen in canonical K$^+$ channels, it has been unclear how to explain

their selectivity in structural terms. To begin to bridge this gap, we again turned to MD simulations. Specifically, we repeated the enhanced-sampling simulations carried out to examine $K^+$ permeation after substituting the KCl solution with NaCl and examined the resulting data analogously (*Figures 2 and 3*). The potential-of-mean-force profile for $Na^+$ is similar to that obtained for $K^+$ in its overall features, including the rate-limiting barrier at the isoleucine constriction (*Figure 2A*). The largest difference between the profiles is the magnitude of the barrier at the isoleucine constriction, which peaks at about 8 kcal/mol relative to bulk for $Na^+$ compared to about 7 kcal/mol for $K^+$. Integration of the potential-of-mean-force profile obtained for $Na^+$ translates into a single-channel conductance of approximately 0.037 pS, that is, the calculated conductance for $Na^+$ is 6-fold smaller than for $K^+$. While available experimental estimates for human indicate TMEM175 a stronger preference for $K^+$, the qualitative agreement between experimental and calculated values suggest the simulations capture the essence of the mechanism of $K^+$ selectivity. As observed for $K^+$, $Na^+$ permeation proceeds via a single-ion mechanism (*Figure 2—figure supplement 2*, *Figure 3B*). Permeation is again opposed by the drastic dehydration of the ion (*Figure 2B*), and favored by its interaction with the electrostatic field created by the protein within the pore (*Figure 2D*), as well as by transient interactions with multiple residues on both cytoplasmic and luminal sides (*Figure 2C*). These stabilizing factors are not identical but appear comparable to those observed for $K^+$. By contrast, the degree of dehydration required for permeation clearly differs. We observe that the depletion in the both the first and second solvation shells, relative to bulk numbers, is significantly smaller for $Na^+$ than for $K^+$. As will be further discussed

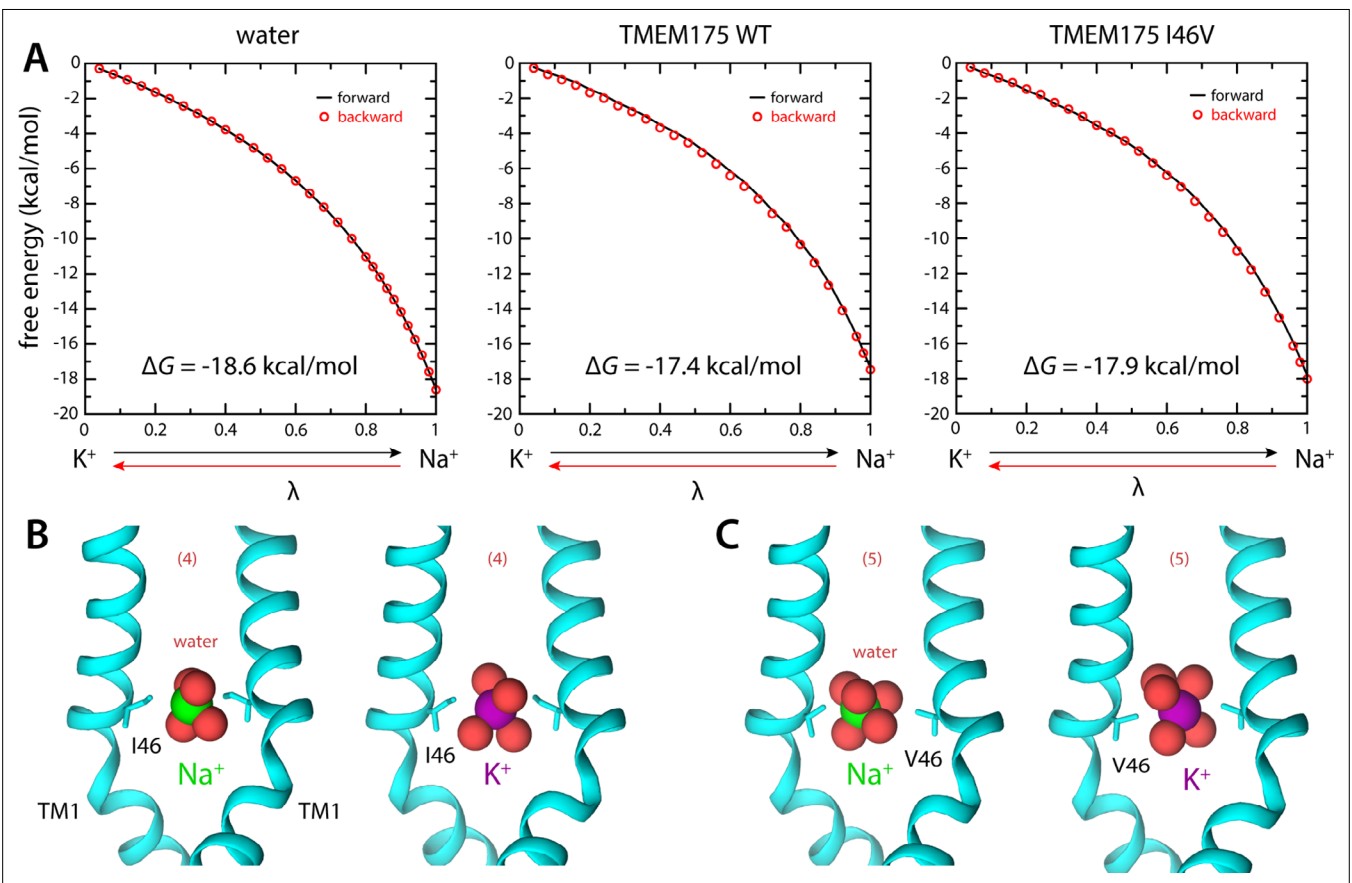

**Figure 4.** Ion selectivity of the isoleucine constriction. (**A**) *Left*, free-energy difference between hydrated $K^+$ and $Na^+$, in bulk water, calculated with molecular dynamics (MD) simulations using the free-energy perturbation (FEP) method. The free energy is plotted as a function of a varying parameter $\lambda$ that determines the size of ion ($\lambda=0$ for $K^+$ and $\lambda=1$ for $Na^+$). Results are shown for a calculation wherein $K^+$ is transformed into $Na^+$ (forward, *black line*) and for another carried out in the opposite direction (backward, *red circles*). *Middle*, analogous free-energy differences, calculated for an ion that is confined within the isoleucine constriction of wild-type TMEM175. *Right*, same as above, for the I46V mutant. Comparison of forward and backward curves provides a metric of the statistical error in these calculations, which is at most 0.1 kcal/mol. (**B**) Snapshots of the MD/FEP simulations at the end of the forward and backward transformations, for wild-type TMEM175 and I46V mutant. For clarity, only TM1 is shown, in cartoon representation, alongside the side chain of I46/V46. Red spheres represent the oxygen atoms of the water molecules in the first ion hydration shell, defined as in *Figure 2*.

below, this relative difference in dehydration energetics likely explains why TMEM175 is only ~30-fold more permeable to K$^+$, even though the bulk water selectivity for Na$^+$ is over a trillion-fold. In other words, like for K$^+$ channels that feature a canonical selectivity filter, it can be said that the narrow pore of TMEM175 favors Na$^+$ over K$^+$, but not as much as bulk water. The channel is thus more permeable to K$^+$ than Na$^+$.

## Role of the isoleucine constriction in ion selectivity and permeation

The isoleucine constriction constitutes the largest free-energy barrier to the permeation of K$^+$ and Na$^+$ in the pore of TMEM175 (*Figure 2*). This narrow constriction forces the ions to shed much of their hydration shells in order to traverse the pore. It was previously demonstrated that mutating both Ile46 and Ile271 to asparagine diminishes the selectivity of TMEM175, indicating that the hydrophobicity of the isoleucine constriction is crucial for selectivity (*Lee et al., 2017*). To determine if the dimensions of the constriction are also important, we used MD simulations to calculate the change in free energy that results when a K$^+$ ion at the isoleucine constriction is exchanged with a Na$^+$ ion, both for wild-type TMEM175 and for a I46V mutant with an expanded isoleucine constriction (Materials and methods). In both cases the exchange results in a free-energy gain (*Figure 4A*), consistent with the notion that Na$^+$ is the favored species at the constriction. However, this gain is smaller than that observed for the exchange of a K$^+$ ion in bulk water with a Na$^+$ ion. As discussed above, it is relative favorability for K$^+$ at the constriction compared to bulk solution that would be the expected result for a K$^+$-selective channel (*Figure 4A*). Specifically, the differential relative to bulk is 1.2 kcal/mol in favor of K$^+$, similar to the ~1 kcal/mol difference in the peak values of the free-energy curves discussed in the previous section (*Figure 2A*). Na$^+$ is however favored to a larger extent in the I46V mutant, whose isoleucine constriction has a similar hydrophobicity but is expanded by the loss of a methyl group from the side chain of Ile46 in each protomer (*Figure 4A*); specifically, the calculations indicate that this mutant is about 2.5-fold (or 0.5 kcal/mol) less K$^+$-selective than the wild-type channel (*Figure 4A*). These results again correlate with the degree of dehydration of each ion, relative to bulk values (*Figure 4B*).

Using whole-cell electrical recordings, we experimentally assessed the effects of expanding the size of the isoleucine constriction on selectivity by measuring the selectivity of Cs$^+$ over Na$^+$ for comparison with the computational analyses. We choose Cs$^+$ rather than K$^+$ because Cs$^+$ blocks most canonical K$^+$ channels, thereby reducing endogenous currents. Nevertheless, whole-cell currents recorded from cells expressing the I46A, I46V, I271A, and I271V mutants in a bi-ionic Cs$^+$/Na$^+$ condition were much smaller than those recorded from wild-type TMEM175 and in some cases similar to currents recorded from non-transfected cells, which hampered accurate determination of ion permeability ratios via measuring reversal potentials ($E_{rev}$) (*Figure 5* and *Figure 5—figure supplement 3*). Thus, isolation of the TMEM175-specific currents from the endogenous HEK293T and background leak was necessary to evaluate the selectivity of the TMEM175 mutants. To this end, we first performed a background subtraction with the TMEM175 inhibitor 4-aminopyridine (4-AP) (*Cang et al., 2015*). However, while 4-AP can effectively inhibit wild-type TMEM175, it was ineffective against the mutants and thus not suitable for isolating the TMEM175-specific currents (*Figure 5—figure supplements 3 and 4*). We therefore repeated the background subtraction experiments with a novel TMEM175 inhibitor that we have developed, which we call AP-6 (*Figure 5* and *Figure 5—figure supplements 1–3*). Following AP-6 background subtraction, I46V and I271V displayed Cs$^+$-selective currents with $E_{rev}$ of −13 and −16 mV, corresponding to Cs$^+$ over Na$^+$ permeability ratios ($P_{Cs/Na}$) of 1.7 and 1.9, respectively (*Figure 5*). Using the same background subtraction approach, we estimated the $P_{Cs/Na}$ of wild-type TMEM175 to be 20 ($E_{rev}$ of −76 mV). This 10-fold reduction in $P_{Cs/Na}$ ion permeation ratio is in line with the computational analyses of the free-energy difference in resulting from K$^+$ to Na$^+$ ion exchange at the isoleucine constriction of I46V and wild type. We were not able to detect any AP-6-sensitive currents from the I46A or I271A mutants either due their insensitivity toward AP-6 or inactivity of the mutant channels. These data demonstrate that increasing the size of the isoleucine constriction leads to a clear reduction in K$^+$ selectivity. Thus, the isoleucine constriction, which is universally conserved among eukaryotic TMEM175 channels, serves as a hydrophobic selectivity filter that is exquisitely sensitive to very subtle changes in its structure.

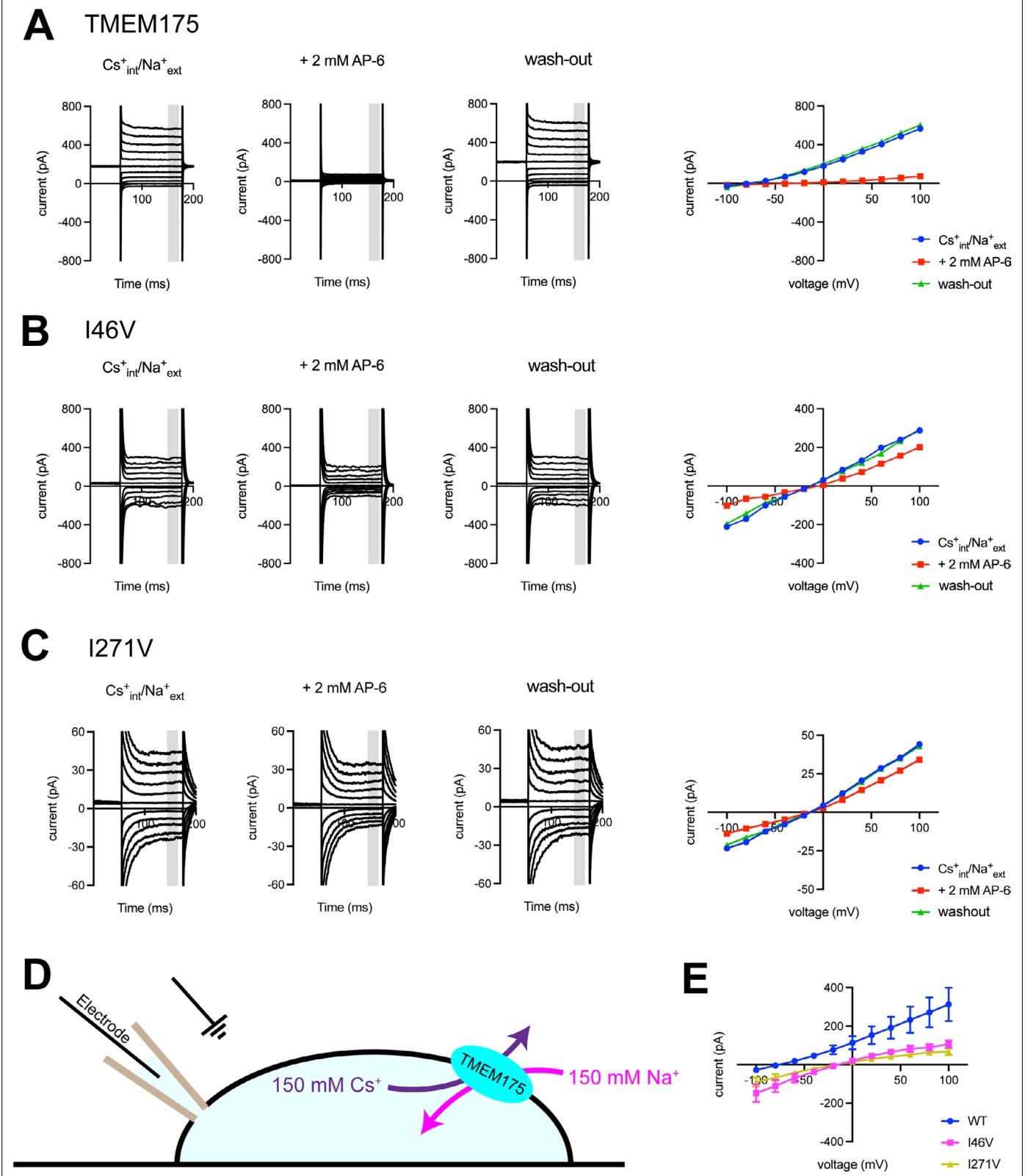

**Figure 5.** Electrophysiological analysis of human transmembrane protein 175 (hTMEM175)-transfected HEK293T cells. Representative whole-cell electrical recordings and current-voltage relationships of hTMEM175-transfected (**A**), I46V-transfected (**B**), and I271V-transfected (**C**) HEK293T cells in the absence and presence of inhibitor AP-6, in bi-ionic conditions of 150 mM Cs$^+$ (intracellular) and 150 mM Na$^+$ (extracellular). Current amplitudes in the current-voltage relationship plots were gained by averaging current values in the shaded time window. (**D**) A schematic of whole-cell patch clamp

*Figure 5 continued*

performed in (**A–C and E**). (**E**) Current-voltage relationships of six independent whole-cell patch clamp recordings of hTMEM175 WT-, I46V-, and I271V-transfected cells in bi-ionic conditions of 150 mM Cs$^+$ (intracellular) and 150 mM Na$^+$ (extracellular).

The online version of this article includes the following source data and figure supplement(s) for figure 5:

**Source data 1.** Source data for *Figure 5*.

**Source data 2.** Source data for *Figure 5*.

**Figure supplement 1.** Characterization of transmembrane protein 175 (TMEM175) inhibition by AP-6.

**Figure supplement 2.** Spectroscopy of AP-6.

**Figure supplement 3.** 4-Aminopyridine (4-AP) and AP-6 sensitivity of transmembrane protein 175 (TMEM175) mutants.

**Figure supplement 4.** 4-Aminopyridine (4-AP) insensitivity of I271V mutant.

## Discussion

Permeation through selective ion channels arises from the interplay between channel-ion interactions, solvent-ion interactions, and in some cases, ion-ion interactions. The energetic cost associated with fully or partially dehydrating an ion such as K$^+$ is so large that permeation would only very rarely occur without favorable interactions with the channel as the ion traverses the conduction pathway. In canonical K$^+$ channels, backbone and side chain oxygen atoms provide a series of direct coordinating interactions that closely resemble those experienced by the ion in bulk water. In addition, these channels are often capable of attracting multiple ions to the pore concurrently. Thus, the cost of dehydration is almost entirely offset by channel-ion and ion-ion interactions, and the resulting free-energy barriers for permeation are small. In contrast, such interactions are lacking for much of the TMEM175 pore, and entirely absent near the isoleucine constriction. Lacking these interactions, it is unlikely that direct protein-ion interactions impart ion selectivity on hTMEM175 channels as was proposed based upon the structural analysis of a TMEM175 homolog from the prokaryote *Marivirga tractuosa* (*Brunner et al., 2020*). K$^+$ ions crossing the pore must nevertheless shed much of their hydration shells; only ~9 of the 31 water molecules in its first two shells remain at the constriction. In addition, ions permeate TMEM175 one at a time, without direct or indirect knock-on interactions. Permeation is however strongly favored by a delocalized electrostatic field across the pore, which offsets much of the cost of ion dehydration, resulting in a free-energy barrier of ~7 kcal/mol (*Figure 2*). While this is a high barrier, it is not insurmountable. Indeed, considering that MD simulations using conventional force fields tend to underestimate ion conduction rates, in some cases up to 10-fold (*Allen et al., 2006*; *Mironenko et al., 2021*), we estimate the unitary conductance of hTMEM175 is 0.1–0.5 pS, which is well within the broad range of experimentally measured permeation rates for other selective cation channels such as ~100 pS for Slo K$^+$ and 9–24 fS for CRAC channels (*Tao et al., 2016*; *Zweifach and Lewis, 1993*). Thus, despite the hydrophobicity of the isoleucine constriction, the unique architecture and amino acid make-up of the TMEM175 pore facilitates ion permeation.

Importantly, the balance between protein-ion interactions and solvent-ion interactions is dependent on the ion type, and so these opposing forces also govern ion selectivity in channels. For example, when rationalizing differences in permeability between K$^+$ and Na$^+$, it is useful to keep in mind that dehydration of Na$^+$ is much more costly than that of K$^+$; the penalty for full dehydration is ~18 kcal/mol greater, rendering it a trillion-fold less probable. However, the observed selectivity of biological K$^+$ channels against Na$^+$ is only 1000-fold or less, implying Na$^+$ establishes much more favorable interactions with the selectivity filters of this kind of K$^+$ channels, as noted in previous studies (*Kim and Allen, 2011*; *Kopec et al., 2018*). In other words, what ultimately determines the specificity of a channel is the sum of the relative cost of dehydration of the competing ions and the differential in their free energy of interaction with the channel. Remarkably, for TMEM175 these favorable protein-ion interactions are imparted by residues near the luminal and cytoplasmic entrances to the pore, quite far from the isoleucine constriction. Thus, we propose that the ion selectivity of TMEM175 primarily results from a differential in the degree of dehydration required for permeation, relative to bulk water. Incidentally, whereas recent studies have revealed differing results for simulations of ion channels performed with different force fields (*Klesse et al., 2020*; *Ocello et al., 2020*), a mechanism that relies on dehydration energetics, rather than specific ion-protein or ion-ion interactions, will be reasonably

described by the force field employed in our simulations to reproduce the relative free energies of hydration (and thus dehydration) of the alkali cations (*Figure 4A*).

Together, our analyses provide a structural and energetic model for the unique mechanisms of ion permeation and selectivity in TMEM175 channels. By combining a narrow hydrophobic sieve with a favorable electrostatic field, hTMEM175 selectively permeates K$^+$ ions across the lysosomal membrane, as was proposed by Lee and colleagues for the TMEM175 homolog from *Chamaesiphon minutus* (*Lee et al., 2017*). As the large conductance Slo1 K$^+$ channel, whose K$^+$ permeation rate is several orders of magnitude higher than TMEM175, has also been observed in lysosomes, it will be important to uncover the specific roles for TMEM175's distinct conduction and selectivity properties in the lysosome and why mutations of TMEM175 can both increase and decrease the likelihood of developing PD (*Wang et al., 2017*; *Zhong et al., 2016*).

# Materials and methods

## Analysis of electron microscopic images

7907 images of TMEM175 in KCl previously acquired in two data sets as 40-frame super-resolution movies (0.544 Å/pixel) using a Gatan K2 (*Oh et al., 2020*) were gain corrected, Fourier cropped by two, and aligned using whole-frame and local motion correction algorithms by Motioncor2 (*Zheng et al., 2017*) (1.088 Å/pixel). Whole-frame CTF parameters were determined using CTFfind 4.1.14 (*Rohou and Grigorieff, 2015*). Particles were automatically selected in Relion 3.0 using templates previously generated from 2D classification, resulting in 4,153,614 particles (*Zivanov et al., 2018*). False-positive selections and contaminants were excluded from the data using multiple rounds of heterogeneous classification in cryoSPARC v3.2 using the open and closed states, as well as several decoy classes generated from noise particles via ab initio reconstruction in cryoSPARC v3.2, resulting in a stack of 636,148 particles (*Punjani et al., 2017*). After Bayesian polishing in Relion and local CTF estimation and higher-order aberration correction in cryoSPARC v3.2, a consensus reconstruction was determined at resolution of 2.8 Å (*Zivanov et al., 2019*). A second round of Bayesian polishing in Relion 3 using a pixel size of 0.85 Å and box size of 384 yielded an improved consensus reconstruction at 2.5 Å. Iterative supervised heterogenous classification using open and closed maps low pass filtered to 6 Å as references resulted in 261,536 particles in the open state and 163,651 particles in the closed state. A 2.45 Å reconstruction of the open state particles and a 2.61 Å reconstruction of the closed state particles were obtained using non-uniform refinement in cryoSPARC v3.2 employing global and per-particle CTF correction. The reconstructions were subjected to density modification using the two unfiltered half-maps with a soft mask in Phenix (*Terwilliger et al., 2019*).

## Model building and coordinate refinement

The structures of open (PDB:6WC9) and closed (PDB:6WCA) hTMEM175 were docked into the density maps in COOT and manually adjusted to fit the density (*Emsley et al., 2010*). Densities corresponding to TM5 and TM6 (residues 174–251) were too poorly ordered and omitted from the model. The final models are composed of residues 30–173 and 254–476. Atomic coordinates were refined against the density modified map using phenix.real_space_refinement with geometric and Ramachandran restraints maintained throughout (*Adams et al., 2010*). Pore radius calculations were performed using HOLE (*Smart et al., 1996*).

## Simulation systems and general specifications

All simulations were calculated with NAMD 2.12 using the CHARMM36 force field for protein and lipids (*Best et al., 2012*; *Klauda et al., 2010*; *Pastor and Mackerell, 2011*; *Phillips et al., 2005*). The simulations were carried out at constant temperature (298 K) and semi-isotropic pressure (1 atm), using periodic boundary conditions and an integration time step of 2 fs. Long-range electrostatic interactions were calculated using PME, with a real-space cut-off of 12 Å. Van der Waals interactions were computed with a Lennard-Jones potential, cut-off at 12 Å with a smooth switching function taking effect at 10 Å.

The simulations were based on the high-resolution cryo-EM structure of the hTMEM175 channel (PDB 6WC9). The specific construct studied includes residues 30–165 and residues 254–476. K$^+$ ions and water molecules originally included in the cryo-EM structure were removed. All ionizable side

chains were set in their default protonation state at pH 7, except for H57, which was protonated on account of its proximity to D279, E282, and D283. The protein construct was embedded in a pre-equilibrated hydrated palmitoyl-oleoyl-phosphatidyl-choline (POPC) lipid bilayer using GRIFFIN, and enclosed in a periodic orthorhombic box of ~100 × 100 × 111 Å³ in size (*Staritzbichler et al., 2011*). The two resulting simulation systems contain 222 POPC lipids, 24,342 water molecules, 49 Cl⁻ ions, and either 43 K⁺ or 43 Na⁺ ions; that is, a salt concentration of 100 mM plus counterions to neutralize the protein net charge. The simulation systems were equilibrated following a staged protocol comprising a series of restrained simulations. The protocol consists of both positional and conformational restraints, gradually weakened over 100 ns. A third simulation system was prepared with 400 mM KCl, by adding 132 K⁺ and 132 Cl⁻ ions to the 100 mM KCl system (replacing water), followed by a 100 ns equilibration.

## Simulation of K⁺ permeation under voltage

To simulate the flow of K⁺ across TMEM175 from the luminal to the cytoplasmic side, a conventional MD trajectory of 1 μs was calculated for the 400 mM KCl condition under a constant electric field perpendicular to the membrane plane, of magnitude $E_z$ = –0.1045 kcal/(mol Å e). Given the average length of the box in this direction ($L_z$ = 110.5 Å), the corresponding TM potential is –500 mV ($\Phi = E_z \times L_z \times 0.0434$ V e⁻¹ (kcal/mol)⁻¹) (*Roux, 2008*).

## Enhanced-sampling simulation of K⁺ and Na⁺ permeation

To induce reversible ion permeation across TMEM175 at 0 mV and 100 mM KCl or 100 mM NaCl, we used the multiple-walker Metadynamics method (*Raiteri et al., 2006*). In Metadynamics, a biasing potential is introduced in an MD simulation to facilitate the exploration of configurational space, as defined by one or more reaction coordinates, also known as collective variables. This biasing potential consists of a series of Gaussian functions that expands as the simulation progresses, to foster the trajectory to visit high free-energy configurations. In the multiple-walker approach, several MD simulations are carried out in parallel, each sampling a different trajectory, but with a shared biasing potential that is also collectively constructed. To study K⁺ permeation, we used eight walkers, each producing a 600 ns trajectory. For Na⁺, we also used eight walkers, 700 ns each.

In biased-sampling simulations of channel permeation, it is common for the position of one or more ions along the pore axis to be used as collective variables. This choice might be reasonable when the number of ions involved in the mechanism is known. It was not known in our case, and hence, the type of collective variable used in these simulations was newly formulated to circumvent a priori assumptions. This variable was implemented in a modified version of PLUMED 1.3 (*Bonomi et al., 2009*). Specifically, the variable is a quantitative descriptor of the proximity between the ions in the simulation system and a virtual center within the pore. More precisely, the variable is:

$$\zeta_{\min} = \frac{\beta}{\log \sum_k exp\{\beta/(|\Delta Z_k|+C)\}} - C \tag{1}$$

where the $k$ index identifies each of the K⁺ or Na⁺ ions in the system and $\Delta Z_k$ is the Cartesian Z-component of the distance between ion $k$ and a center-of-mass defined by a group of protein atoms. $C$ is a positive constant (set to 2 Å) that avoids numerical instabilities when $\Delta Z_k \sim 0$ and $\beta$ is a smoothing parameter (set to 100 Å). To successfully induce ion permeation events, we used two variables of the kind specified by *Equation 1*, namely $\zeta_{\min}^A$ and $\zeta_{\min}^B$, each defined in reference to a center of atoms. Center A is defined by the Cα atoms of residues 44–47 and 268–271, while center B is defined by the Cα atoms of residues 45–48 and 269–272. Centers A and B are therefore along the pore axis, separated by 1 Å and flanking the isoleucine constriction. The Gaussian functions used to gradually construct the biasing potentials applied to $\zeta_{\min}^A$ and $\zeta_{\min}^B$ had a width 0.25 Å and were added in 4 ps intervals. To reduce systematic errors near the boundary $\zeta_{\min} = 0$, reflected and inverted Gaussians were added beyond this boundary for both variables as previously described (*Crespo et al., 2010*). In the simulations of K⁺ permeation, the Gaussians height was gradually raised from 0.0035 to 0.007 kcal/mol in the first 30 ns of simulation, and gradually diminished back to 0.0035 kcal/mol after 100 ns. For Na⁺, the Gaussians height was gradually raised from 0.0035 to 0.007 kcal/mol in the first 50 ns of simulation and diminished back to 0.0035 kcal/mol after 250 ns.

## Derivation of free energies and other quantitative descriptors from biased-sampling trajectories

A post hoc reweighting procedure was used to derive unbiased averages and histograms from the MD trajectories enhanced by the Metadynamics biasing potential (*Marinelli et al., 2009*). In this approach, a time average of the Metadynamics biasing potential is calculated after it becomes approximately stationary across the range of $(\zeta_{min}^A, \zeta_{min}^B)$ values of interest. (Here, the last 400 ns of simulation for $K^+$ and the last 600 ns of simulation for $Na^+$.) This effective potential, $\bar{V}\left(\zeta_{min}^A, \zeta_{min}^B\right)$, is then used to adjust the statistical weight of each of the simulation snapshots considered in the analysis. Specifically, the statistical weight of a given snapshot $X_i$ is:

$$w\left(\mathbf{X_i}\right) = \frac{\exp\{\bar{V}(\zeta_{min}^A(\mathbf{X_i}), \zeta_{min}^B(\mathbf{X_i}))/k_B T\}}{\sum_j \exp\{\bar{V}(\zeta_{min}^A(\mathbf{X_j}), \zeta_{min}^B(\mathbf{X_j}))/k_B T\}} \tag{2}$$

where $k_B$ is the Boltzmann constant and $T$ is the temperature. That is, snapshots $X_i$ that fall in easily accessible regions of $(\zeta_{min}^A, \zeta_{min}^B)$ where the accumulated Metadynamics bias is large are given a greater statistical weight, while those in unfavorable regions where the accumulated bias is less are also assigned a smaller weight. The effective potential for each simulation snapshot was derived using a 2D discretization across the space of $(\zeta_{min}^A, \zeta_{min}^B)$, wherein each snapshot is assigned to a grid point and an associated value of $\bar{V}$ using a recently developed tool for free-energy analysis (*Marinelli and Faraldo-Gómez, 2021*).

Following this approach, it is straightforward to derive unbiased estimates of the average ion occupancy along the length of the channel pore, and thus of the corresponding free-energy landscape. Specifically, let us define this length by an axis connecting the abovementioned centers A and B (which fluctuates with the channel but is approximately perpendicular to the membrane plane). For each ion $k$ in the simulation system, let us also define $z_k\left(\mathbf{X_i}\right)$ and $R_k\left(\mathbf{X_i}\right)$ as the projection of the ion coordinates along the pore axis and the distance to that axis, respectively, for a given snapshot $\mathbf{X_i}$. If we divide the pore axis into a series of intervals or bins, the ion occupancy of a given bin α centered in zα is:

$$\rho\left(z_\alpha\right) = \sum_i \sum_{k \in (\alpha, R_0)} w\left(\mathbf{X_i}\right) \tag{3}$$

where the $i$ index again denotes each trajectory snapshot included in the analysis, and the sum over $k$ is restricted to ions whose $z_k(\mathbf{X_i})$ coordinate falls into bin $\alpha$ and that are also proximal to the pore axis ($R_k(X_i) \leq R_0 = 10$ Å). It follows that the potential of mean force (or free-energy profile) associated with this occupancy distribution along zα will be:

$$F\left(z_\alpha\right) = -k_B T \log \rho\left(z_\alpha\right) = -k_B T \log \sum_i \sum_{k \in \alpha} w\left(\mathbf{X_i}\right) + C' \tag{4}$$

where $C'$ is a constant arbitrarily selected so that $F \sim 0$ in the bulk, where the profile is nearly flat.

We followed same reweighting approach to derive unbiased estimates for other descriptors, namely average ion coordination numbers and average ion-protein electrostatic interaction energies (definitions provided below). Specifically, the mean value of either of these observables at a given position along the pore axis is:

$$\bar{O}\left(z_\alpha\right) = \frac{1}{\rho\left(z_\alpha\right)} \sum_i \sum_{k \in (\alpha, R_0)} O_k\left(\mathbf{X_i}\right) w\left(\mathbf{X_i}\right) \tag{5}$$

where $i$ again denotes a trajectory snapshot, $O_k\left(\mathbf{X_i}\right)$ is the descriptor of interest of ion $k$, in that particular snapshot, and the sum over $k$ is again restricted to ions within the pore and within bin α.

## Calculation of ion coordination numbers

To quantify the coordination of $K^+$ or $Na^+$ by either water or protein oxygen atoms as a function of the ion position along the pore axis, we evaluated the following function of the ion-oxygen distances for each simulation snapshot $\mathbf{X_i}$ and each ion $k$ in the system:

$$S_k\left(\mathbf{X_i}\right) = \sum_j \frac{1 - \left(r_{jk}(\mathbf{X_i})/r_0\right)^{100}}{1 - \left(r_{jk}(\mathbf{X_i})/r_0\right)^{200}} \tag{6}$$

where the $j$ index denotes all possible coordinating oxygens, and $r_{jk}(X_i)$ denotes their distance to ion $k$ in snapshot $X_i$. Note this function is virtually identical to a step function cut-off at distance equal to $r_0$. For computational expediency, this evaluation was carried out with PLUMED 1.3 (*Bonomi et al., 2009*). For the evaluation of the number of water molecules in the first hydration shell, $r_0$ was set to 3.5 Å for K$^+$ and to 3.2 Å for Na$^+$, which in each case corresponds to the position of the first minimum of the ion-oxygen radial distribution functions (RDF) in bulk water (data not shown). The same values of $r_0$ were used for calculating the coordination with protein oxygens. Following the same criteria, the number of water molecules in the first and second shells was calculated by setting $r_0$ to 6.0 and 5.7 Å, respectively, as these values reflect the position of the second minimum of the RDF in each case. The two exponents of the step function in *Equation 6* were set to reproduce the integral of the RDF in bulk water within $r_0$.

## Calculation of ion-protein electrostatic interaction energies

To estimate the electrostatic interaction energy between each K$^+$ or Na$^+$ ion and the channel, we used a continuum electrostatic model based on the linearized form of Poisson's equation. That is, for each ion $k$ and snapshot $X_i$, we evaluated the electrostatic potential $\Phi_k$ generated at the position of the ion by the atomic charges in the protein structure ($q_p$), in the context of a heterogenous dielectric-constant distribution defined by the protein ($\varepsilon_p = 2$ or $1$), an implicit membrane surrounding the protein ($\varepsilon_m = 2$), the bulk water solvent ($\varepsilon_s = 80$), the water solvent inside the pore ($\varepsilon_{s'} = 40$), and the ion in question ($\varepsilon_k = 2$ or $1$). (Note that both charge and dielectric-constant distributions are a function of $X_i$, as is the position of the ion.) The interaction between this electrostatic potential and the ion is given by:

$$E_k(X_i) = q_k \Phi_k(q_p, \varepsilon_p, \varepsilon_m, \varepsilon_s, \varepsilon_{s'}, \varepsilon_k; X_i) \tag{7}$$

These calculations were carried out with the PBEQ numerical solver implemented in CHARMM c44b1 (*Brooks et al., 2009*). The set of atomic charges employed are those in the CHARMM36 force field; the set of atomic radii used to define the dielectric boundaries derives from an existing optimization for continuum-electrostatic calculations based on the CHARMM27 force field (*Best et al., 2012*; *Nina et al., 1997*). Electrostatic potential, charge, and dielectric-constant distributions were discretized on a lattice of $150 \times 150 \times 150$ Å$^3$, with lattice-point spacing of 1 Å. The thickness of the implicit membrane was 28.8 Å. The low-dielectric span of pore water was defined by a cylinder of radius 18 Å and height 28.8 Å. The protein surface was defined using the REEN method, using a probe radius of 1 Å.

## Calculation of ion conductance

To obtain an approximate estimate of the K$^+$ and Na$^+$ conductance of TMEM175 channel, we used the theoretical formulation proposed by *Zhu and Hummer, 2012*, wherein the conductance $\gamma$ is inferred from the free-energy ($F(z)$) and diffusion ($D(z)$) profiles as a function of the ion position along the pore axis:

$$\gamma = \frac{q^2 CS}{k_B T} \left[ \int_{z_1}^{z_2} \exp\left\{ \frac{F(z)}{k_B T} \right\} \frac{1}{D(z)} dz \right]^{-1} \tag{8}$$

where $q$ (=1) is the charge of the permeant ion, $C$ is the bulk ion concentration (100 mM), and $S$ is the cross-sectional area (314 Å$^2$) of the cylindrical region considered to evaluate *Equations 3–5*. As a first approximation, we assumed the diffusion profile be flat, and used $D(z) \sim 2 \times 10^{-9}$ m$^2$/s for K$^+$ and $D(z) \sim 1.5 \times 10^{-9}$ m$^2$/s for Na$^+$, following previous simulation studies (*Zhou et al., 2017*; *Zhu and Hummer, 2012*). Note that $\gamma$ depends linearly with $D(z)$, while the dependence on $F(z)$ is exponential; thus, while it is very plausible that the diffusion constant will vary by at least a factor of 2 as the ion traverses the pore (*Zhou et al., 2017*), the order of magnitude of the conductance is largely set by the features of the free-energy profile.

## Calculation of free energies of selectivity

To evaluate free-energy differences between K$^+$ and Na$^+$ states, whether in bulk water or at the isoleucine constriction within the TMEM175 pore, we used the free-energy perturbation method

as implemented in NAMD 2.12. To ascertain the magnitude of sampling errors, all transformations were carried out in the forward ($K^+$ to $Na^+$) and backward ($K^+$ to $Na^+$) directions. The transformations were carried out in 26 steps; each step included a 50 ps equilibration, excluded from analysis, followed by 500 ps of averaging time. For bulk water, the coupling parameter lambda was varied in increments of 0.04 in the [0, 0.8] interval and in increments of 0.02 in the [0.8, 1] interval. For the protein calculations, lambda was changed in increments of 0.04 in the [0, 0.96] interval and in increments of 0.02 in the [0.96, 1] interval. To ensure that the $K^+$ and $Na^+$ ions involved in the transformation are in the same position, the distance between the two particles is restrained to zero using a harmonic potential of force constant 2425.58 kcal/(mol $Å^2$). In the protein calculations, the ion(s) are confined to remain within the isoleucine constriction using a flat-bottom distance restraint, defined relative to the center-of-mass of the backbone atoms of two groups of residues, namely 44–45, 269–270 (first group) and 47–49, 272–274 (second group). These two groups also define an axis; the restraint acts on the distance between the ion and the center-of-mass, as projected on that axis. The confining potential is flat-bottomed, permitting fluctuations of ±0.1 Å in the distance, but further deviations are suppressed with a harmonic function of force constant 100 kcal/(mol $Å^2$).

## Electrophysiological analysis

Electrophysiological recordings of TMEM175 constructs were performed in HEK293T cells (ATCC CRL-3216). HEK293T cells were recently purchased from ATCC and are negative for mycoplasma contamination. HEK293T cells cultured in DMEM supplemented with 10% FBS. To transfer cells in single dishes, cells were detached by trypsin treatment. The detached cells were transferred to poly-Lys-treated 35 mm single dishes (FluoroDish, World Precision Instruments) and incubated overnight at 37°C in fresh media. Cells in a single dish were transfected with 1.25 µg of c-term EGFP tagged hTMEM175 plasmid using 3.75 µg of PEI 25 k (Polysciences, Inc). Electrophysiological recordings were performed 48–72 hr after transfection. Prior to recording, media was replaced with a bath solution containing 145 mM Na-methanesulfonate (MS), 5 mM NaCl, 1 mM $MgCl_2$, 1 mM $CaCl_2$, 10 mM HEPES/Tris pH 7.4. Ten-cm-long borosilicate glasses were pulled and fire polished (Sutter instrument). Glass pipettes of resistances between 3 and 10 MΩ were filled with a pipette solution containing 150 mM Cs-MS, 5 mM $MgCl_2$, 10 mM EGTA, 10 mM HEPES/Tris pH 7.4, and GΩ seals were formed after gentle suction. The recordings were performed in whole-cell patch clamp configuration using the following protocol: from a holding potential of 0 mV, the voltage was stepped to voltages between −100 and +100 mV, in 20 mV increments. To measure currents reduced by 4-AP or AP-6, bath solutions were perfused with a solution containing 1 mM 4-AP or 2 mM AP-6. DMSO concentration was maintained as 0.33% during the entire experiment. The currents were recorded using Axon Digidata 1550B digitizer and Clampex 10.6 (Molecular Devices, LLC) and analyzed using AxoGraph X 1.7.6 (AxoGraph Scientific). Each experiment was performed in a unique cell. To determine the $IC_{50}$ of 6-AP, the dose-response curve for TMEM175 current at +100 mV in the presence of AP-6 was fit with the following equation:

$$I = I_{\min} + \left(I_{\max} - I_{\min}\right) \times \frac{IC_{50}}{IC_{50} + \left[\text{AP-6}\right]} \tag{9}$$

where $I$ is the normalized current, $I_{\max}$ and $I_{\min}$ are maximum and minimum normalized current, respectively. [AP-6] is the concentration of AP-6 perfused.

## General chemistry

Chemical reagents and materials were purchased from Sigma-Aldrich, Thermo-Fisher, and TCI, and used without further purifications. DCM (dichloromethane), MeOH, and *n*-hexane for column chromatography and recrystallization were used for HPLC grade without additional purifications. Thin layer chromatography (TLC) analysis was performed for reaction monitoring on the pre-coated silica gel 60 F254 glass plates. Both starting materials and the desired product were checked by UV light (254 nm). Flash column chromatography was carried out on silica gel (400–630 mesh) to separate the target molecule.

## Synthesis of compound 1 2,2'-(1,3-phenylene)bis(pyridin-4-amine) (AP-6)

AP-6 was prepared through Suzuki-Miyaura cross-coupling reaction with palladium catalysis. In a round-bottom flask with a magnetic bar, a mixture of 2-bromopyridin-4-amine (5 mmol, 865 mg), 1,3-phenylenediboronic acid (0.75 equiv., 622 mg), $K_2CO_3$ (2 equiv., 1.38 g), and $Pd(OAc)_2$ (7 mol%, 78.6 mg) were dissolved in $H_2O$:EtOH solution mixture (8 mL: 32 mL), and the solution was stirred at 100°C for 24 hr under air. After completion (monitored by TLC), the reaction mixture was filtered through a Celite (after cooling to room temperature), and then the solid on the filter was washed with EtOAc. The mixture was added to brine and extracted with EtOAc for three times. The combined organic layer was dried over $MgSO_4$, filtered, and concentrated under reduced pressure. The residue was purified by chromatography in silica gel (DCM/MeOH), and purified again with recrystallization (n-hexane/DCM) to give the desired product.

Proton nuclear magnetic resonance ($^1$H NMR) spectra were recorded on Bruker AVANCE 500 (500 MHz). In addition, $^{13}$C{$^1$H} NMR was measured on the same machine (125 MHz), and the spectra was fully decoupled with proton by broad band decoupling. Chemical shifts for NMR were quoted in parts per million referenced to the appropriate solvent peak (DMSO in DMSO-$d_6$). The abbreviation codes were adopted to describe $^1$H NMR peak patterns; d = doublet, t = triplet, and br = broad. Coupling constants, $J$, were displayed in Hertz unit (Hz). Infrared (IR) spectra were recorded on Bruker Alpha FT-IR spectrometer. High-resolution mass spectra were acquired on a high-resolution Q-TOF mass spectrometer (ionization mode: ESI).

## Acknowledgements

We thank the MSKCC HPC group for assistance with data processing and the members of the labs for comments on the manuscript. We also thank Dr Rahul Banerjee for early simulation studies of closed state TMEM175 not included in this manuscript. This work was supported by NIH-NCI Cancer Center Support Grant P30 CA008748, NIGMS R01-GM141553 (RKH), the Josie Robertson Investigators Program (RKH), the Searle Scholars Program (RKH), the NRF Global PhD. Fellowship program funded by the Republic of Korea Ministry of Education 2019H1A2A1076014 (JL) and the Division of Intramural Research of NHLBI-NIH (JDFG). Computational resources were in part provided by the NIH supercomputing center (Biowulf).

## Additional information

### Competing interests

Fabrizio Marinelli: José D Faraldo-Gómez: The other authors declare that no competing interests exist.

### Funding

| Funder | Grant reference number | Author |
| --- | --- | --- |
| National Cancer Institute | P30 CA008748 | Richard K Hite |
| National Institute of General Medical Sciences | R01-GM141553 | Richard K Hite |
| Josie Robertson Investigators Program | | Richard K Hite |
| Searle Scholars Program | | Richard K Hite |
| Ministry of Education | 2019H1A2A1076014 | Jooyeon Lee |
| National Heart, Lung, and Blood Institute | Division of Intramural Research | José D Faraldo-Gómez |

The funders had no role in study design, data collection and interpretation, or the decision to submit the work for publication.

## Author contributions
SeCheol Oh, Conceptualization, Data curation, Formal analysis, Investigation, Methodology, Resources, Validation, Visualization, Writing – original draft, Writing – review and editing; Fabrizio Marinelli, Wenchang Zhou, Conceptualization, Data curation, Formal analysis, Investigation, Methodology, Resources, Software, Validation, Visualization, Writing – original draft, Writing – review and editing; Jooyeon Lee, Ho Jeong Choi, Investigation, Methodology, Resources, Validation; Min Kim, Formal analysis, Funding acquisition, Investigation, Methodology, Project administration, Resources, Supervision, Validation, Visualization, Writing – review and editing; José D Faraldo-Gómez, Conceptualization, Data curation, Formal analysis, Funding acquisition, Investigation, Methodology, Project administration, Resources, Software, Supervision, Validation, Visualization, Writing – original draft, Writing – review and editing; Richard K Hite, Conceptualization, Data curation, Formal analysis, Funding acquisition, Investigation, Methodology, Project administration, Resources, Supervision, Validation, Visualization, Writing – original draft, Writing – review and editing

## Author ORCIDs
SeCheol Oh (iD) http://orcid.org/0000-0002-1685-5922
Fabrizio Marinelli (iD) http://orcid.org/0000-0003-0044-6718
Wenchang Zhou (iD) http://orcid.org/0000-0003-0397-1032
José D Faraldo-Gómez (iD) http://orcid.org/0000-0001-7224-7676
Richard K Hite (iD) http://orcid.org/0000-0003-0496-0669

## Decision letter and Author response
Decision letter https://doi.org/10.7554/eLife.75122.sa1
Author response https://doi.org/10.7554/eLife.75122.sa2

# Additional files

## Supplementary files
• Transparent reporting form

## Data availability
Cryo-EM maps and atomic coordinates have been deposited with the EMDB and PDB under accession codes EMD-26626 and PDB 7UNL for open TMEM175 and codes EMD-26627 and PDB 7UNM for closed TMEM175. Source data have been provided for Figure 5.

The following datasets were generated:

| Author(s) | Year | Dataset title | Dataset URL | Database and Identifier |
|---|---|---|---|---|
| Oh S, Hite RK | 2022 | Open state TMEM175 model | https://www.rcsb.org/structure/7UNL | RCSB Protein Data Bank, 7UNL |
| Oh S, Hite RK | 2022 | Open state TMEM175 map | https://www.emdataresource.org/EMD-26626 | EMDataResource, EMD-26626 |
| Oh S, Hite RK | 2022 | Closed state TMEM175 model | https://www.rcsb.org/structure/7UNM | RCSB Protein Data Bank, 7UNM |
| Oh S, Hite RK | 2022 | Closed state TMEM175 map | https://www.emdataresource.org/EMD-26627 | EMDataResource, EMD-26627 |

The following previously published dataset was used:

| Author(s) | Year | Dataset title | Dataset URL | Database and Identifier |
|---|---|---|---|---|
| Oh S, Hite RK | 2020 | Open state TMEM175 model | https://www.rcsb.org/structure/6WC9 | RCSB Protein Data Bank, 6WC9 |

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

# Appendix 1

**Appendix 1—table 1.** Cryo-EM data acquisition, reconstruction, and model refinement statistics.

| | TMEM175 Open | TMEM175 Closed |
|---|---|---|
| **Cryo-EM acquisition and processing** | | |
| EMDB accession # | EMD-26626 | EMD-26627 |
| Magnification | 22,500× | 22,500× |
| Voltage (kV) | 300 | 300 |
| Total electron exposure (e⁻/ Å²) | 61 | 61 |
| Exposure time (s) | 8 | 8 |
| Defocus range (μM) | −1.0 to −2.5 | −1.0 to −2.5 |
| Pixel size (Å) | 1.088 | 1.088 |
| Final pixel size (Å) | 0.85 | 0.85 |
| Symmetry imposed | C2 | C2 |
| Initial particles | 4,153,614 | 4,153,614 |
| Final particles | 261,536 | 163,651 |
| Resolution (masked, Å) | 2.45 | 2.61 |
| Density modified CC (0.5, Å) | 2.43 | 2.65 |
| **Model refinement** | | |
| PDB ID | 7UNL | 7UNM |
| Model resolution (Å) | 2.48/1.96 | 2.88/2.49 |
| FSC threshold | 0.50/0.143 | 0.50/0.143 |
| Model refinement resolution | 300–2.4 | 300–2.6 |
| **RMS deviations** | | |
| Bond length (Å) | 0.003 | 0.002 |
| Bond angle (°) | 0.559 | 0.433 |
| **Ramachandran plot** | | |
| Favored (%) | 98.9 | 99.45 |
| Allowed (%) | 1.1 | 0.55 |
| Disallowed (%) | 0 | 0 |
| Rotamer outliers (%) | 1.94 | 0 |
| **Validation** | | |
| MolProbity score | 1.22 | 1.13 |
| Clashscore | 2.24 | 3.44 |

