## [Editor Report]

This manuscript explores the mechanisms of permeation and selectivity in the unusual potassium-selective ion channel TMEM175, which lacks a canonical selectivity filter. The study is led by molecular dynamics simulations and free energy calculations, complemented by a cryoEM analysis and electrophysiological recordings. The authors propose a novel, single ion-based mechanism of permeation, together with a partial dehydration-driven selectivity mechanism. This study will appeal to readers interested in the structure and function of ion channels and in molecular mechanisms of ion translocation.

---

## [Decision Letter]

**Decision letter after peer review:**

Thank you for submitting your article "Differential ion dehydration energetics explains selectivity in the non-canonical lysosomal K^+^ channel TMEM175" for consideration by *eLife*. Your article has been reviewed by 3 peer reviewers, and the evaluation has been overseen by a Reviewing Editor and Richard Aldrich as the Senior Editor. The reviewers have opted to remain anonymous.

Essential revisions:

1) Improved justification of methodological choices.

2) A characterization of the uncertainties on the free energies.

3) A demonstration that a reasonable choice of a different force field would give similar results.

4) A more thorough discussion of agreement (or lack thereof) with other (e.g., experimental) data.

*Reviewer #1 (Recommendations for the authors):*

1) The Introduction reads as if the authors were the first and only group of determining the structure of a TMEM175 channel. While they were the first for the human isoform, two bacterial isoforms have been structurally characterized and were found to have almost identical pore structure. Those works also formulated hypotheses about the mechanisms of selectivity and gating; both identified the isoleucine constriction. One of these articles is mentioned in passing (Lee et al., 2017, and it's not becoming clear that this was a structural work), while the other one is not cited at all in the entire manuscript (Brunner et al. 2020, *eLife* 9, the authors cite a preprint of this work in their 2020 article). Also, the only previous theoretical work on the TMEM175 (Rao et al. 2018 in Faraday Discuss. 209) is not cited. With only a low single-digit number papers available on the structure and mechanism of TMEM175, they should be given credit. The same applies for the discussion; a discussion of these new results in relation to the literature is lacking.

2) The section "Improved cryo-EM structures of hTMRM175" is a follow-up to the previous paper (Oh et al., 2020), reporting on a refined on the crystal structure data.

a) It is unclear to me how the new and improved structural model relates to that one from last year. The authors merely mention that it is similar.

b) Were there any changes in the structure or ion position due to the new refinement, or were the now better resolved densities in the expected places? Are the ion binding sites K1-K4 in the same position as in the previous work? Since the notation K1-K4 was dropped (why?), it is very hard for the reader to tell by comparison.

3) Sections "Energetics and mechanism of K^+^ permeation" and "Differentials in ion dehydration relative to bulk explain selectivity for K^+^ over Na+"

a) It is not mentioned whether the minima in the energy surfaces or the binding sites observed in the time traces coincide with the ion binding sites in the cryoEM structure. This is important in the light that the authors specifically point out that they made no a priori assumptions about this.

b) The comparison of K^+^ and Na^+^ comes a bit short. Although the degree of water depletion is different for K^+^ and Na+, they end up at the same number of 4 remaining waters. Coordination by the protein is also very similar and it stays unclear whether these two points are relevant for selectivity. Permeation simulations for both K^+^ and Na^+^ were performed, but the numerical results (number of permeation events, estimated conductivity) are only given for K^+^, so the comparison of both ions remains somewhat vague.

c) How appropriate is it to use a constant electric field for a channel with an hourglass pore, where most of the field drops likely over the narrow constriction?

d) The Supplement 2 to Figure 2 is not very helpful. It is very crowded. I was able to pick out some instances where one line seems to cross the entire gap in the middle, which I presume is an ion transition. But I see those both for K^+^ and Na+. In Figure 3, Na^+^ is also seen to permeate. How are the statistics of permeation events for K^+^ and Na+? What is the estimated conduction for 100 mM Na+? I am aware that permeation does not equal selectivity, but I am curious as to why this basic information was left out.

4) Section "Role of the isoleucine constriction in ion selectivity and permeation"

a) Brunner et al. (2020) showed experimentally that not only the isoleucine, but also a stack of polar residues strongly contribute to selectivity in both the bacterial and human isoforms. They also formulated a hypothesis on the gating mechanism. Does this contradict or complement the results of this paper? The hypothetical open state modelled by Rao et al. (2018) is also not mentioned.

b) The new blocker AP-6 seems to be indeed slightly more effective at blocking the mutants than 4-AP. But it seems do drop out of thin air. Neither the design rationale nor whether it is specific to TMEM175 is discussed.

*Reviewer #2 (Recommendations for the authors):*

1) Error estimates for the calculated free energies and free energy profiles should be provided to see if found differences of ca. 1 kcal/mol are statistically significant. Similarly, the applied field simulations show two permeation events, a rather low number, that should also be treated with care.

2) The differences of ca. 1 kcal/mol are around the accuracy of current force fields for free energy predictions, and the authors are using only one force field (CHARMM) for a somewhat difficult target (an ion in a hydrophobic environment) that was not explicitly parameterized in this force field. Given that in the field of canonical potassium channels force field inaccuracies led to some 20 years long discussion about the permeation mechanisms, I'd suggest to either include an investigation with another force field, or tone down the conclusions (for example that this is a 'proposed permeation mechanism for TMEM175') and discuss potential issues with the current methodology. A recent paper showing a likely too hydrophobic character of the TMEM175 cavity in nonpolarizable force fields (Lynch et al., ACS Nano 2021) will be also of interest here.

3) The authors state in the abstract that the channel is 'capable of permeating K^+^ ions at the expected rate' and later (page 12) their estimated conductance of human TMEM175 is 0.1-0.5 pS, which the authors comment as "well within the broad range of experimentally measured permeation rates". Can the authors actually provide some numbers here? For example, the bacterial TMEM175 shows a conductance of ca. 70 pS (Brunner et al. *eLife* 2020) and Slo1 K^+^ ca. 100 pS (Tao et al. Nature 2017).

4) For clarity, I suggest calculating the conductance for sodium from PMFs the way it was done for potassium; then, compare the K^+^/Na^+^ permeability ratios obtained from MD with the experiment. If possible, the error estimates for the permeability ratios should be included as well.

5) Throughout the manuscript, the authors sometimes use the thermodynamic description (i.e. differences in binding free energies between K^+^ and Na^+^ ions at a given site) to explain experimentally measured ion selectivity (i.e. permeation rates), that naturally also contains the kinetic part (height of free energy barriers). For example: 'it is useful to keep in mind that dehydration of Na + is much more costly than that of K + ; the penalty for full dehydration is ~18 kcal/mol greater, rendering it a trillion-fold less probable. However, the observed selectivity of biological K^+^channels against Na+is only 1000-fold or less, implying Na^+^ establishes much more favorable interactions with the selectivity filters of this kind of K^+^ channels.'

I'd be careful to directly connect Na^+^ dehydration penalty with experimentally observed permeation rates, and make a clear distinction in the manuscript between kinetic and thermodynamic selectivity derived from MD. Moreover 'Na + establishes much more favorable interactions with the selectivity filter of this kind of K^+^ channels' is not only implied but actually shown before, see free energy calculations in Kim et al., PNAS 2011 and Kopec et al. Nat. Chem. 2018.

6) The authors seem to ignore several interesting insights into ion selectivity of TMEM175 channels reported recently by Brunner et al. *eLife* 2020 – the paper is not even cited. It would be very beneficial for the whole field to include a discussion on how the authors' mechanism and findings agree (or not) with those of Brunner et al.

7) line 78 – table S1 is absent in the manuscript.

8) line 129 – "we detected no evidence of a multi-ion process (Figure 2—figure supplement 2)" – I assume it refers to the number of ions present at the same time at the level of the constriction, however it is not clear from the figure. I suggest defining a "multi-ion" permeation process in the text and put coordinates of SF on Figure 2—figure supplement 2 to make it clear.

9) line 203 – "specifically, the calculations indicate this mutant is about 2-fold less K^+^ selective than the wild-type channel" – it is not calculated in the manuscript at the moment.

10) lines 177-181 – "We observe that the depletion in the both the first and second solvation shells, relative to bulk numbers, is significantly smaller for Na+than for K^+^. As will be further discussed below, this relative difference in dehydration energetics likely explains whyTMEM175 is only~30-fold more permeable to K^+^, even though the bulk water selectivity for Na+is over a trillion-fold." – it may not be clear how the number of water molecules lost during dehydration affects selectivity by itself (rather than the free energy difference between the processes of going from the bulk to the constriction). If it's the case, it should be explained more clearly.

11) Figure 5 Supplementary Figure 3 – the points for the raw currents before adding AP-6 are much more scattered than before 4-AP. What could be the reason for this behavior?

6. line 492 – "To measure currents reduced by AP-6, bath solutions were perfused" – the method for measuring the currents reduced by 4-AP are not described.

12) The starting structure for the simulations is stated to be 6WC9, the previously published structure of the open TMEM175, even though structures with higher resolution were obtained in this study. The reasoning behind using a lower-resolution structure should be provided.

13) Even though the positions of ions as a function of time are shown in Figure 2 Supplementary Figure 2, it may not be enough to estimate the convergence of metadynamics simulations. Can the authors provide an example of the time dependence of the CVs, or the deposited potential, or some other suitable measure for convergence as well?

14) As another control for the metadynamics simulations, would it be possible to run MD of TMEM175 with the potassium ions and water molecules not removed from the initial structure? It could show if there are some energy minima not resolved by metadynamics, and if those ions/water molecules have any effect on the overall behavior of TMEM175. Also, would you expect any side effects from not modelling residues 164-253?

*Reviewer #3 (Recommendations for the authors):*

1) While the refined static structure of the open channel is discussed in detail, the conformational dynamics of the pore is not: was there any conformational isomerization of side chains? What is the extent of fluctuations in pore radius and relative helix arrangement? Do the apparent kink and tilt of the pore helices fluctuate?

2) There seems to be a discrepancy between the fully-hydrated state of ions in the cryoEM densities (lines 95-97, Figure 1 Suppl. Figure 2) and their partial dehydration in the simulations (Figure 2). This is not a trivial point, since the major finding of the paper is that selective K^+^ permeation arises from ion (de)solvation effects.

3) Systematic error analysis of the simulation results would strengthen the confidence in the numerical agreement noted above.

4) A broader discussion of the general significance of the findings, including the role of ion desolvation effects and the novelty that a hydrophobic locus controls both gating and selectivity in K^+^ channels and ion channels in general, would be welcome.

5) The introduction could provide a bit more background. What is the biological function of this protein? Only dysfunction is mentioned.

6) Please specify which atoms were used for the analysis shown in Figure 2 Suppl. Figure 1 (Calpha atoms, etc).

7) It would be useful to computational biophysicists if the authors could clarify the rationale for specific methodological choices. In particular, what are the considerations that presided over the choice of force field? Same question for the collective variable used in the metadynamics simulations.

8) Likewise, why did the authors use the particular functional form of Equation 6 to compute the ion coordination number rather than a simple cut-off distance? Please provide references for that choice if appropriate. Also, please note that brackets are missing in the summations in Eq. 6.

9) Error estimates should be provided systematically. In the current manuscript, they are sometimes vague or non-existent. In particular:

– Please provide data for estimating the convergence for metadynamics; and generally, error estimates as appropriate for all the numerical results.

– The penultimate sentence in the caption of Figure 2 is unclear: "gray profiles represent the same quantity shown in black/blue calculated using only the first or second half of the simulations data."

10) Please provide a reference for variations in the value of the diffusion constant (see lines 453-454).

11) I would recommend including a schematic figure of a thermodynamic cycle to the Discussion for clarity and to highlight the consistency between the two different pathways followed for the DeltaDeltaG calculation (see "Strengths" in public review above).

12) The statement in lines 236-237 seems to imply that the balance of channel-ion, ion-ion, and water-ion interactions controls selective ion permeation. The authors may consider including the effect of channel-water (and arguably also water-water and channel-channel) interactions for the sake of completeness and generality.

[Editors' note: further revisions were suggested prior to acceptance, as described below.]

Thank you for resubmitting your work entitled "Differential ion dehydration energetics explains selectivity in the non-canonical lysosomal K^+^ channel TMEM175" for further consideration by *eLife*. Your revised article has been evaluated by Richard Aldrich (Senior Editor) and a Reviewing Editor.

The manuscript has been improved but there are some remaining issues that need to be addressed, as outlined below. We anticipate acceptance after these changes have been implemented.

Please address the comments by reviewers 1 and 3, as detailed below.

In addition, regarding the justification of the force field choice, we wish to highlight that claiming that default CHARMM ion parameters from 1994 (Beglov and Roux, 1994) reproduce the relative free-energies of hydration (and thus dehydration) of the alkali cations 'with excellent accuracy' somewhat invalidates next 20 years of ion force field development. When addressing the issue in the article's text, as requested by reviewer 3, please ensure the comment is accurate and well-substantiated, and do not forget to mention possible shortcomings of the chosen force field.

*Reviewer #1 (Recommendations for the authors):*

I thank the authors for their careful revision of the manuscript. Most of my previous comments have been addressed adequately, except for these two:

Regarding the position of the ion binding sites in the different figures created by different methods: Noting the residues in Figure 2 is certainly helpful (as has already been done in the original manuscript). However, indicating K1-K5 directly as I suggested in my initial review would spare the reader to flip back and forth the figures and comparing each single residue manually. There is still no indication of the binding sites in the time traces of the MD simulations, my request "All figures where this applies: please indicate the location of the binding sites from the structural data" has been ignored by the authors without any comment.

I still strongly suggest amending the figures accordingly as this would make them more accessible.

This comment of mine has maybe been overlooked by the authors: What is the meaning of the numbers in the bracket in Figure 3A? I presume the number of water molecules?

*Reviewer #2 (Recommendations for the authors):*

All points have been adequately addressed. I recommend the paper for publication.

*Reviewer #3 (Recommendations for the authors):*

The authors have addressed most of the comments appropriately.

In their response concerning the choice of forcefield, they write: "It could be convincingly argued that a mechanism that relies on dehydration energetics, rather than specific ion-protein interactions, will be reasonably described by this forcefield, as it is in fact parametrized, with excellent accuracy, to reproduce the relative free energies of hydration (and thus dehydration) of the alkali cations." That is the kind of justification that I was hoping for. Please make that convincing argument in the text, including references as needed.

---

## [Author Response]

Essential revisions:1) Improved justification of methodological choices.

We thank the reviewers for their questions. We have included the clarifications suggested by the reviewers below, and when appropriate, in the revised manuscript.

2) A characterization of the uncertainties on the free energies.

As requested, we now have clarified how we had estimated the uncertainties of the free energies. For the Metadynamics calculations described in Figure 2, we had plotted free-energy profiles based on either the first 50% or the second 50% of all trajectory data. Comparison of these two independent profiles provides a straightforward metric of convergence and hence statistical error. The statistical uncertainties of the free-energy barrier for permeation of K^+^ and Na^+^ are 0.14 kcal/mol and 0.22 kcal/mol, respectively. These data confirm that the statistical error of the calculations is smaller than the difference in the free energy between K^+^ and Na+.

In Figure 4, which reports on the FEP calculations, we had superposed the results for the forward and backward transformations. By comparing these plots, we can obtain a simple metric of convergence for this type of calculation. As in Figure 2., the statistical errors in these calculations (0.1 kcal/mol) are much smaller than any of the computed free-energy differences. The excellent agreement between Metadynamics calculations and FEP yield, despite being radically different methodologies, is further evidence that the statistical error in these calculations is minimal. We thank the reviewers for encouraging us to clarify these important analyses.

3) A demonstration that a reasonable choice of a different force field would give similar results.

We thank the editor and reviewers for their suggestion. However, for the reasons enumerated below we disagree that replicating the simulations using a different forcefield should be essential for publication of this work:

1. The reviews do not articulate what specific concern with the forcefield used in our simulations could, in the reviewers’ opinion, significantly alter the conclusions of our study. Reviewer #2 points that out that “an ion in a hydrophobic environment […] was not explicitly parameterized in [the CHARMM36] force field”. Indeed, no general forcefield is explicitly parameterized in any one specific context. The premise of all general forcefields used in molecular simulations of biomolecular systems is precisely that they will yield plausible results even when they are not explicitly parameterized for the structure being examined or for a specific problem of interest. Requiring that studies employ explicitly parameterized forcefields, which are rare, would greatly limit the use of simulations for analyzing biomolecular systems. Problem-dependent corrections to a general forcefield can and have been of course developed to improve the accuracy of specific elements when evidence exists that these elements are not only incorrectly captured but also critical. Two examples from our own laboratory are a comprehensive examination of anion-protein interactions (Orabi et al., JCTC 2021) as well as sodium- and calcium-protein interactions (Liao et al., NSMB 2016). Analogous corrections have been used to examine channels wherein K^+^ ions are directly multi-coordinated by carbonyls or carboxyl groups (Noskov et al., Nature 2001 among others). For the specific process examined in our study, however, we at this time have no reason to question the approximate validity of the CHARMM36 forcefield. Indeed, it could be convincingly argued that a mechanism that relies on dehydration energetics, rather than specific ion-protein interactions, will be reasonably described by this forcefield, as it is in fact parameterized, with excellent accuracy, to reproduce the relative free-energies of hydration (and thus dehydration) of the alkali cations.

2. This request appears to imply that the plausibility of a simulation study is questionable unless its conclusions can be replicated using different forcefields. Are all forcefields equally predictive, for any given problem? There is no reason to assume that they are. Then why should a result be the same for different forcefields? And how does one determine which forcefields must be compared, and how many? Should one compare forcefields that are similar in essence, like CHARMM and AMBER? Or would the critique then be that they are too similar? Or should one select forcefields that are fundamentally different – like fixed-charge vs polarizable – despite the fact they might be at radically different stages of development and testing? For example, our own recent attempts to calculate differences in the free energy of interaction between ions and proteins using the Drude/CHARMM36 polarizable forcefield, with NAMD, revealed critical flaws in the implementation of the algorithm, as this calculation had apparently not been attempted before. This situation illustrates the difficulties and potential pitfalls of this type of comparison. It is unquestionable that simulation studies are inherently hypothetical, regardless of forcefield, or the myriad other assumptions and simplifications adopted in these studies. Reviewer #2 points out that “in the field of canonical potassium channels force field inaccuracies led to some 20 years long discussion about the permeation mechanisms”. Well, that is the fundamental nature of the scientific enterprise; the same could be said about most important problems, whether examined with experimental or computational techniques. The way to deal with the uncertainties inherent to the scientific enterprise is careful presentation and interpretation. In the case of simulation work, it needs to be understood that it does not result in definitive answers to a given problem, but rather a working hypothesis based on a specific model. What makes a computational study merit publication is not that it can be replicated against all possible models – but that it presents results that are statistically significant and reproducible for a given set of conditions, obtained with state-of-the-art quantitative methods, and most importantly, that it provides a plausible – but necessarily tentative – interpretation to experiment, or a verifiable prediction.

3. It is not the editorial policy of *eLife*, or other reputable journals, to generically require that simulation studies be replicated with different forcefields in order to merit publication, without a specific concern or justification. As articulated above, there are good reasons that explain why that is the case.

4) A more thorough discussion of agreement (or lack thereof) with other (e.g., experimental) data.

We thank the reviewers for encouraging us to provide additional discussion of our work. We now add several additional sections to the text. In the introduction, we describe the previous structural studies on prokaryotic TMEM175 channels and highlight some of the important structural and functional differences between prokaryotic TMEM175 and human TMEM175 channels. In the discussion we state that our results are consistent with the suggestion of Lee and colleagues that a hydrophobic constriction imparts ion selectivity on TMEM175 channels but disagree with Brunner and colleagues who suggest that the hydrophobic constrictions serve exclusively as channel gates while direct protein-ion interactions impart ion selectivity.

In our previous work, we observed layers of ordered water molecules immediately above and below the isoleucine constriction in the putative open state that were partially coordinated by the side chains of polar residues including Ser45 and Thr274. Based on this observation, we mutated Ser45 conservatively to threonine and non-conservatively to alanine. We similarly mutated Thr274 to serine and valine. For both residues, the non-conservative mutations resulted in a loss of channel activity, while the conservative mutations had a lesser effect. The S45T and T274S mutations also resulted in a slight reduction in ion selectivity, which may have resulted from greater influence of the non-selective endogenous currents on the smaller currents from the mutant channels. We therefore proposed that Ser45 and Thr274 are critical for ion permeation due to their role in stabilizing water molecules near the isoleucine constriction; thus, the permeating ion retains its first hydration shell except when traversing the constriction.

In agreement with our work, Brunner and colleagues proposed that Ile46/Ile271 form the channel gate. However, they did not observe a similar loss of activity for the S45A mutant or a T49A/T274A double-mutant, but rather a loss of selectivity for K^+^ over Na^+^. We are currently unclear of the origin of the conflicting results between the groups regarding the effect of mutating Ser45 and Thr49/Thr274. However, as we observe only minimal interactions between Ser45, Thr49 or Thr274 and the K^+^ ions in the simulations (Figure 3C), the computational results are consistent with our previous conclusion that Ser45 or Thr49/Thr274 do not have a strong role in establishing ion selectivity. Future studies will be required to uncover the basis of the disagreements.

Reviewer #1 (Recommendations for the authors):1) The Introduction reads as if the authors were the first and only group of determining the structure of a TMEM175 channel. While they were the first for the human isoform, two bacterial isoforms have been structurally characterized and were found to have almost identical pore structure. Those works also formulated hypotheses about the mechanisms of selectivity and gating; both identified the isoleucine constriction. One of these articles is mentioned in passing (Lee et al., 2017, and it's not becoming clear that this was a structural work), while the other one is not cited at all in the entire manuscript (Brunner et al. 2020, eLife 9, the authors cite a preprint of this work in their 2020 article). Also, the only previous theoretical work on the TMEM175 (Rao et al. 2018 in Faraday Discuss. 209) is not cited. With only a low single-digit number papers available on the structure and mechanism of TMEM175, they should be given credit. The same applies for the discussion; a discussion of these new results in relation to the literature is lacking.

We thank the reviewer for their suggestion. We have added a section to the introduction describing the previous structural studies on prokaryotic TMEM175 channels and the structural and functional differences between human TMEM175 and prokaryotic TMEM175 homologs, focusing on their different pore structures, oligomeric state and selectivity profiles. We have also added a section to the discussion comparing our results with the findings described by other groups.

2) The section "Improved cryo-EM structures of hTMRM175" is a follow-up to the previous paper (Oh et al., 2020), reporting on a refined on the crystal structure data.a) It is unclear to me how the new and improved structural model relates to that one from last year. The authors merely mention that it is similar.b) Were there any changes in the structure or ion position due to the new refinement, or were the now better resolved densities in the expected places? Are the ion binding sites K1-K4 in the same position as in the previous work? Since the notation K1-K4 was dropped (why?), it is very hard for the reader to tell by comparison.

We thank the reviewer for their suggestions. The RMSD between the protein atoms of the updated open structure and the previously published open structure (PDB:6WC9) is 0.3 Å and the and ion-binding sites are all located within 0.6 Å of their previously assigned positions. The RMSD between the protein atoms of the updated closed structure and the previously published closed structure (PDB:6WCA) is 0.4 Å and the ion-binding sites are all located within 1 Å of their previously assigned positions.

We have added labels to Figure 1 for the ion binding sites.

3) Sections "Energetics and mechanism of K^+^ permeation" and "Differentials in ion dehydration relative to bulk explain selectivity for K^+^ over Na+"a) It is not mentioned whether the minima in the energy surfaces or the binding sites observed in the time traces coincide with the ion binding sites in the cryoEM structure. This is important in the light that the authors specifically point out that they made no a priori assumptions about this.

We thank the reviewer for their suggestion and have revised the manuscript to clarify this question and highlight the similarities of these two orthogonal approaches to identify ion binding sites in the pore of TMEM175. The position of residues lining the permeation pathway is indicated alongside the density in the revised version of Figure 1 and alongside the free-energy profile calculated for K^+^ in Figure 2. There is very good agreement between these observations. Densities near S38 and A42 on the cytoplasmic side and near V50 on the luminal side appear as metastable states in the calculated free-energy curve. Densities right below and above the central constriction also appear as shoulders in the free-energy profile.

b) The comparison of K^+^ and Na^+^ comes a bit short. Although the degree of water depletion is different for K^+^ and Na+, they end up at the same number of 4 remaining waters. Coordination by the protein is also very similar and it stays unclear whether these two points are relevant for selectivity. Permeation simulations for both K^+^ and Na^+^ were performed, but the numerical results (number of permeation events, estimated conductivity) are only given for K^+^, so the comparison of both ions remains somewhat vague.

We thank the reviewer for highlighting this important finding of our work. It is precisely because K^+^ and Na^+^ interact similarly with the protein (unlike other K^+^ or Na^+^ channels) that we argue that the selectivity of this channel primarily stems from differences in their dehydration energetics. That in both cases the hydration shells are depleted down to the same number of water molecules at and near the constriction in fact implies different energetic penalties relative to bulk water – which as we show are sufficient to explain the measured difference in permeability. A direct metric of these differences is provided by the free-energy calculation in which one ion is replaced by the other while residing at the constriction, relative to the same exchange in bulk water – see Figure 4. In summary, the different degree of depletion is not an inconsequential observation; it is actually a very important factor and one of the unique features of this channel.

c) How appropriate is it to use a constant electric field for a channel with an hourglass pore, where most of the field drops likely over the narrow constriction?

The physical quantity that drops over a narrow constriction in a transmembrane pore is not the applied electric field – it is the electrostatic potential. Specifically, it is the potential resulting not only from the applied field but also from the response of the molecular system to this applied field. Thus, an ion diffusing in bulk water or within water-accessible regions of a wide pore will experience essentially no change in the surrounding electrostatic potential, even if the applied electric field is constant across the same region. The reason is that water is highly dielectric, and the applied electric field is largely countered (or screened) by the field that results from the preferential orientation of water molecules in response to the applied field. By contrast, as a pore becomes narrower the orientational freedom of water and thus its screening effect are diminished, and so the net electric field within is non-zero; an ion traveling from a wider region into this narrower region will thus experience a change in electrostatic potential. How to correctly simulate transmembrane potentials has been discussed at length in the existing literature. See for example Roux, Biophys J 2008, or Gumbart et al., BBA Biomembranes 2012. These and other studies “demonstrate that the constant-field method is a simple and valid approach for accounting for the membrane potential in molecular dynamics studies of biomolecular systems”.

d) The Supplement 2 to Figure 2 is not very helpful. It is very crowded. I was able to pick out some instances where one line seems to cross the entire gap in the middle, which I presume is an ion transition. But I see those both for K^+^ and Na+.

Trajectories that cross the central gap are indeed crossing events. The figure captions have been revised to clarify this question. That we see crossings for both K^+^ and Na^+^ is expected as the channel is permeable to both ions. It should be however noted that the free-energy profiles shown in Figure 2 are based on the totality of these data, not only crossing events. It is for that reason that we have opted to preserve these plots in the revised version. For clarity the revised manuscript now includes two movies depicting ion permeation events (induced by voltage).

In Figure 3, Na^+^ is also seen to permeate. How are the statistics of permeation events for K^+^ and Na+? What is the estimated conduction for 100 mM Na+? I am aware that permeation does not equal selectivity, but I am curious as to why this basic information was left out.

We did not simulate Na^+^ permeation under voltage. Na^+^ permeation was examined only at 0 mV, using enhanced-sampling simulations – see Figure 2 and related figure supplements. From this data, the estimated conduction for Na^+^ at 100 mV is 0.04 pS, which is 5-fold smaller than what we estimate for K^+^. Both estimates are indicated in the revised manuscript.

4) Section "Role of the isoleucine constriction in ion selectivity and permeation"a) Brunner et al. (2020) showed experimentally that not only the isoleucine, but also a stack of polar residues strongly contribute to selectivity in both the bacterial and human isoforms. They also formulated a hypothesis on the gating mechanism. Does this contradict or complement the results of this paper? The hypothetical open state modelled by Rao et al. (2018) is also not mentioned.

In our previous work, we observed layers of water molecules immediately above and below the isoleucine constriction that were partially coordinated by the side chains of polar residues including Ser45 and Thr274. Based on this observation, we hypothesized that the water molecules may contribute to ion permeation. To test the hypothesis, we mutated Ser45 conservatively to threonine and non-conservatively to alanine. We similarly mutated Thr274 to serine and valine. For both residues, the non-conservative mutations resulted in a loss of channel activity, while the conservative mutations had a lesser effect. The S45T and T274S mutations also resulted in a slight reduction in ion selectivity, which may have resulted from increased influence of the endogenous currents on the smaller currents. We therefore proposed that Ser45 and Thr274 are critical for ion permeation due to their role in stabilizing water molecules near the isoleucine constriction.

In agreement with our work, Brunner and colleagues proposed that Ile46/Ile271 form the channel gate. However, they did not observe a similar loss of activity for the S45A mutant or a T49A/T274A double-mutant, but rather a loss of selectivity for K^+^ over Na^+^. We are currently unclear of the origin of the conflicting results between the groups regarding the effect of mutating Ser45 and Thr49/Thr274. However, as we observe only minimal interactions between Ser45, Thr49 or Thr274 and the K^+^ ions in the simulations (Figure 3C), the computational results are consistent with our previous conclusion that Ser45 or Thr49/Thr274 do not have a strong role in establishing ion selectivity.

The hypothetical open state modelled by Rao and colleagues is a model of a prokaryotic TMEM175 homolog. As the pore structures between human and prokaryotic TMEM175 channels differ due to sequence divergence and oligomeric state, our work on human TMEM175 does not inform on the structure of the open state of a prokaryotic TMEM175 channel. Moreover, as the selectivity profiles of human TMEM175 and prokaryotic TMEM175 are distinct and the mechanisms of gating in prokaryotic TMEM175 channels are unknown, our analyses do not inform on their ion selectivity and gating mechanisms. Due to these fundamental differences between human and prokaryotic TMEM175 channels, we would prefer not to speculate on prokaryotic TMEM175 channels.

b) The new blocker AP-6 seems to be indeed slightly more effective at blocking the mutants than 4-AP. But it seems do drop out of thin air. Neither the design rationale nor whether it is specific to TMEM175 is discussed.

We are preparing a manuscript describing our studies into describing novel TMEM175 inhibitors, including AP-6 in which the rationale and specificity of AP-6 will be characterized.

Reviewer #2 (Recommendations for the authors):1) Error estimates for the calculated free energies and free energy profiles should be provided to see if found differences of ca. 1 kcal/mol are statistically significant. Similarly, the applied field simulations show two permeation events, a rather low number, that should also be treated with care.

Please see our response to the editor’s summary above. We have revised the captions of Figure 2 and Figure 4 to describe how our error calculations were performed. The mean difference between PMF profiles obtained with the first or second half of the trajectory data for K^+^ is 0.14 kcal/mol; for Na^+^, it is 0.22 kcal/mol. The difference between forward and backward FEP transformations is 0.1 kcal/mol. Differences of ca. 1 kcal/mol between K^+^ and Na^+^ are therefore statistically significant.

2) The differences of ca. 1 kcal/mol are around the accuracy of current force fields for free energy predictions, and the authors are using only one force field (CHARMM) for a somewhat difficult target (an ion in a hydrophobic environment) that was not explicitly parameterized in this force field. Given that in the field of canonical potassium channels force field inaccuracies led to some 20 years long discussion about the permeation mechanisms, I'd suggest to either include an investigation with another force field, or tone down the conclusions (for example that this is a 'proposed permeation mechanism for TMEM175') and discuss potential issues with the current methodology. A recent paper showing a likely too hydrophobic character of the TMEM175 cavity in nonpolarizable force fields (Lynch et al., ACS Nano 2021) will be also of interest here.

Please see our response to the editor’s summary above. We have revised the text to underscore we propose a mechanism of selectivity and that the mechanism is supported by three approaches, but that future studies will be helpful in evaluating our proposed mechanism.

3) The authors state in the abstract that the channel is 'capable of permeating K^+^ ions at the expected rate' and later (page 12) their estimated conductance of human TMEM175 is 0.1-0.5 pS, which the authors comment as "well within the broad range of experimentally measured permeation rates". Can the authors actually provide some numbers here? For example, the bacterial TMEM175 shows a conductance of ca. 70 pS (Brunner et al. eLife 2020) and Slo1 K^+^ ca. 100 pS (Tao et al. Nature 2017).

We thank the reviewer for their suggestion and have revised the text to include several examples covering the wide range of conductance values measured for cation channels.

4) For clarity, I suggest calculating the conductance for sodium from PMFs the way it was done for potassium; then, compare the K^+^/Na^+^ permeability ratios obtained from MD with the experiment. If possible, the error estimates for the permeability ratios should be included as well.

We thank the reviewer for their suggestion. When examined at 0 mV using enhanced-sampling simulations, the estimated conduction for Na^+^ at 100 mV is 0.04 pS. The calculated K^+^/Na^+^ permeability ratio is thus 5-fold, which is similar to measurements that others have reported (10-36). We have included both estimates in the revised manuscript.

5) Throughout the manuscript, the authors sometimes use the thermodynamic description (i.e. differences in binding free energies between K^+^ and Na^+^ ions at a given site) to explain experimentally measured ion selectivity (i.e. permeation rates), that naturally also contains the kinetic part (height of free energy barriers). For example: 'it is useful to keep in mind that dehydration of Na + is much more costly than that of K + ; the penalty for full dehydration is ~18 kcal/mol greater, rendering it a trillion-fold less probable. However, the observed selectivity of biological K^+^channels against Na+is only 1000-fold or less, implying Na^+^ establishes much more favorable interactions with the selectivity filters of this kind of K^+^ channels.'I'd be careful to directly connect Na^+^ dehydration penalty with experimentally observed permeation rates, and make a clear distinction in the manuscript between kinetic and thermodynamic selectivity derived from MD.

We thank the reviewer for highlighting this important distinction. We designed and carried out two very different calculations precisely to evaluate alternative potential explanations. PMF profiles derived from Metadynamics trajectories show that for both ions the rate permeation is controlled by a single free-energy barrier; the height of that barrier is greater for Na^+^ than K^+^, explaining why the channel is more permeable to K^+^. That difference is recapitulated by the FEP calculations, which by construction only probe the barrier top (relative to bulk), and which in this case reflect mostly differences in dehydration energetics. Taken together, these results support our working hypothesis that differences in dehydration energetics explain the selectivity of this channel.

Moreover 'Na + establishes much more favorable interactions with the selectivity filter of this kind of K^+^ channels' is not only implied but actually shown before, see free energy calculations in Kim et al., PNAS 2011 and Kopec et al. Nat. Chem. 2018.

These references have been added to the bibliography.

6) The authors seem to ignore several interesting insights into ion selectivity of TMEM175 channels reported recently by Brunner et al. eLife 2020 – the paper is not even cited. It would be very beneficial for the whole field to include a discussion on how the authors' mechanism and findings agree (or not) with those of Brunner et al.

Please see our response to reviewer 1 above regarding the work from Brunner and colleagues. While our work disagrees with the studies described by Brunner and colleagues, our work supports the hypothesis of Lee and colleagues that a narrow hydrophobic sieve and favorable electric field can together facilitate the selective permeation of K^+^ ions.

7) line 78 – table S1 is absent in the manuscript.

Table S1 was inadvertently left out of the original submission and has been added the revised version.

8) line 129 – "we detected no evidence of a multi-ion process (Figure 2—figure supplement 2)" – I assume it refers to the number of ions present at the same time at the level of the constriction, however it is not clear from the figure. I suggest defining a "multi-ion" permeation process in the text and put coordinates of SF on Figure 2—figure supplement 2 to make it clear.

We thank the reviewer for this suggestion. The statement has been clarified.

9) line 203 – "specifically, the calculations indicate this mutant is about 2-fold less K^+^ selective than the wild-type channel" – it is not calculated in the manuscript at the moment.

The stated 2-fold factor directly stems from the calculated free-energy differences in Figure 4: p = log (DG/k_B_T). This has been clarified.

10) lines 177-181 – "We observe that the depletion in the both the first and second solvation shells, relative to bulk numbers, is significantly smaller for Na+than for K^+^. As will be further discussed below, this relative difference in dehydration energetics likely explains whyTMEM175 is only~30-fold more permeable to K^+^, even though the bulk water selectivity for Na+is over a trillion-fold." – it may not be clear how the number of water molecules lost during dehydration affects selectivity by itself (rather than the free energy difference between the processes of going from the bulk to the constriction). If it's the case, it should be explained more clearly.

We thank the reviewer and have clarified this point in the manuscript. In Figure 4, we show that the change in free energy for a K^+^ ion moving from bulk to the constriction is 1.2 kcal/mol less costly than moving a Na^+^ ion from the bulk to the constriction. In Figure 2, we show that both ions enter the constriction with four water molecules in their primary hydration shell, and another four in the second shell. Thus, while the absolute number of water molecules lost during dehydration is greater for K^+^ (3 for K^+^ and 2 for Na^+^ in the first shell; and about 20 for K^+^ and 16+ for Na^+^ in the second shell), it is actually more costly for Na^+^ to achieve the same dehydration state at the constriction than it is for K^+^, because in the bulk Na^+^ interacts with its hydration shells much more strongly than K^+^ (by about 18 kcal/mol).

11) Figure 5 Supplementary Figure 3 – the points for the raw currents before adding AP-6 are much more scattered than before 4-AP. What could be the reason for this behavior?6. line 492 – "To measure currents reduced by AP-6, bath solutions were perfused" – the method for measuring the currents reduced by 4-AP are not described.

We thank the reviewer for noting that we did not describe how the 4-AP currents were measured. We have revised the methods section to include a description. The difference between the current levels prior to AP-6 are due to the variation in channel expression from cell to cell. All cells were prepared and recorded from in an identical fashion.

12) The starting structure for the simulations is stated to be 6WC9, the previously published structure of the open TMEM175, even though structures with higher resolution were obtained in this study. The reasoning behind using a lower-resolution structure should be provided.

The simulations were initiated prior to obtaining the newer high-resolution structure of TMEM175 in the open state. As the structures are essentially identical (RMSD = 0.3 Å), we continued to use 6WC9 for consistency.

13) Even though the positions of ions as a function of time are shown in Figure 2 Supplementary Figure 2, it may not be enough to estimate the convergence of metadynamics simulations. Can the authors provide an example of the time dependence of the CVs, or the deposited potential, or some other suitable measure for convergence as well?

It was not our intention for the data in Figure 2 Supplementary Figure 2 to serve as a convergence metric. As discussed above, to evaluate the convergence of the Metadynamics data we compared PMF profiles derived with either the first or the second half of all trajectory data – these profiles were shown in Figure 2. We have clarified how we evaluated convergence in the revised figure captions.

14) As another control for the metadynamics simulations, would it be possible to run MD of TMEM175 with the potassium ions and water molecules not removed from the initial structure? It could show if there are some energy minima not resolved by metadynamics, and if those ions/water molecules have any effect on the overall behavior of TMEM175. Also, would you expect any side effects from not modelling residues 164-253?

We thank the reviewer for their suggestions and we will consider them in future studies.

Reviewer #3 (Recommendations for the authors):1) While the refined static structure of the open channel is discussed in detail, the conformational dynamics of the pore is not: was there any conformational isomerization of side chains? What is the extent of fluctuations in pore radius and relative helix arrangement? Do the apparent kink and tilt of the pore helices fluctuate?

We thank the reviewer for highlighting the close correspondence between the conformation of the pore in the structures and during the simulations. As illustrated in Figure 2 Supplementary Figure 1 the structural fluctuations of the pore are very limited. No noticeable changes in tilt or kink angles in any of the helices flanking the pore were detected, or large changes in the pore radius. The only significant change concurrent with ion permeation across the constriction is the rotation of the terminal CD atom in the sidechain of Ile 271 around to the CB-CG bond. We have noted the minimal changes in the text and two supplementary movies with different time-resolution have been added to revised manuscript to document these observations.

2) There seems to be a discrepancy between the fully-hydrated state of ions in the cryoEM densities (lines 95-97, Figure 1 Suppl. Figure 2) and their partial dehydration in the simulations (Figure 2). This is not a trivial point, since the major finding of the paper is that selective K^+^ permeation arises from ion (de)solvation effects.

We thank the reviewer for their comment and have revised the manuscript to note that the ordered ions in the cryo-EM structures are not fully hydrated, extending their description from the original manuscript, which stated that “The binding site on the cytoplasmic site of the constriction is coordinated by four ordered water molecules […], while the site on the luminal side is coordinated by four water molecules […].” The partial dehydration of the ions is also shown in Figure 1 S2. These data thus reflect a depletion of the ion hydration shells, as observed in simulation and we have remarked on this finding in the revised manuscript.

3) Systematic error analysis of the simulation results would strengthen the confidence in the numerical agreement noted above.

Please see our responses to Editor’s summary and Reviewer #2.

4) A broader discussion of the general significance of the findings, including the role of ion desolvation effects and the novelty that a hydrophobic locus controls both gating and selectivity in K^+^ channels and ion channels in general, would be welcome.

We thank the reviewer for their suggestion. However, discussion on the unique (to our knowledge) of a hydrophobic constriction participating in ion selectivity as well as serving as the channel gate was included in our previous work. As the gating of TMEM175 remains poorly understood at the molecular level and this study is focused on the selectivity of the channel, such a discussion would be largely redundant if repeated in the current study. Therefore, we would prefer to focus our discussion on the findings that we present in the current study.

5) The introduction could provide a bit more background. What is the biological function of this protein? Only dysfunction is mentioned.

We thank the reviewer for their suggestion and have extended our introduction of the physiological roles of TMEM175.

6) Please specify which atoms were used for the analysis shown in Figure 2 Suppl. Figure 1 (Calpha atoms, etc).

Corrected.

7) It would be useful to computational biophysicists if the authors could clarify the rationale for specific methodological choices. In particular, what are the considerations that presided over the choice of force field? Same question for the collective variable used in the metadynamics simulations.

We thank the reviewer for the opportunity to describe our rationale. As indicated in the original version of the manuscript we designed a collective variable that does not presuppose a mechanism of permeation. In biased-sampling simulations of ion channel permeation it is common to use the coordinate of one or more ions along the pore axis as collective variables. This choice might be reasonable when the number of ions involved in the mechanism is known. It was not known in our case, and hence the variable we designed is a metric of the proximity between any ion in the simulation system and a virtual center within the pore; by using two of these centers, one at either side of the constriction, we were able to foster sampling of the length of the pore and the central barrier, as well as a sufficient number of crossing events. This reasoning has been further clarified in the revised version of the manuscript. Regarding the choice of forcefield, we opted for what we believe to be the best compromise between accuracy and performance, based on our own experience as well as the specific nature of the questions addressed in this study.

8) Likewise, why did the authors use the particular functional form of Equation 6 to compute the ion coordination number rather than a simple cut-off distance? Please provide references for that choice if appropriate. Also, please note that brackets are missing in the summations in Eq. 6.

We chose this form for computational convenience. As explained in the Methods section, to rigorously derive the profiles shown in Figure 2B from trajectories calculated under an applied bias requires calculation of coordination numbers for every ion in every snapshot in every replica trajectory. We found that PLUMED, which features the function defined in Equation 6 as a collective variable, was the most efficient tool to carry out this analysis. Note that the large exponents imply that the function is essentially the same as a square step function cut off at a certain distance. This point has been clarified in the revised manuscript.

9) Error estimates should be provided systematically. In the current manuscript, they are sometimes vague or non-existent. In particular:– Please provide data for estimating the convergence for metadynamics; and generally, error estimates as appropriate for all the numerical results.– The penultimate sentence in the caption of Figure 2 is unclear: "gray profiles represent the same quantity shown in black/blue calculated using only the first or second half of the simulations data."

Please see our responses to Editor’s summary and Reviewer #2 – as well as the revised captions of Figures 2 and 4.

10) Please provide a reference for variations in the value of the diffusion constant (see lines 453-454).

The reference has been added.

11) I would recommend including a schematic figure of a thermodynamic cycle to the Discussion for clarity and to highlight the consistency between the two different pathways followed for the DeltaDeltaG calculation (see "Strengths" in public review above).

We appreciate the reviewer’s suggestion but believe this figure is not essential.

12) The statement in lines 236-237 seems to imply that the balance of channel-ion, ion-ion, and water-ion interactions controls selective ion permeation. The authors may consider including the effect of channel-water (and arguably also water-water and channel-channel) interactions for the sake of completeness and generality.

We have no data that indicates that differentials in channel-water interactions are a determining factor for the ion selectivity of this channel.

[Editors' note: further revisions were suggested prior to acceptance, as described below.]

The manuscript has been improved but there are some remaining issues that need to be addressed, as outlined below. We anticipate acceptance after these changes have been implemented.Please address the comments by reviewers 1 and 3, as detailed below.In addition, regarding the justification of the force field choice, we wish to highlight that claiming that default CHARMM ion parameters from 1994 (Beglov and Roux, 1994) reproduce the relative free-energies of hydration (and thus dehydration) of the alkali cations 'with excellent accuracy' somewhat invalidates next 20 years of ion force field development. When addressing the issue in the article's text, as requested by reviewer 3, please ensure the comment is accurate and well-substantiated, and do not forget to mention possible shortcomings of the chosen force field.

We agree that there have been important advances in force field development. As requested, our revised text now includes a discussion on our chosen force field along with citations of recent studies comparing force fields for simulations of ion channels.

Reviewer #1 (Recommendations for the authors):I thank the authors for their careful revision of the manuscript. Most of my previous comments have been addressed adequately, except for these two:

We thank the reviewer for their comments that have greatly improved our manuscript. We have addressed these two points in the revised version.

Regarding the position of the ion binding sites in the different figures created by different methods: Noting the residues in Figure 2 is certainly helpful (as has already been done in the original manuscript). However, indicating K1-K5 directly as I suggested in my initial review would spare the reader to flip back and forth the figures and comparing each single residue manually. There is still no indication of the binding sites in the time traces of the MD simulations, my request "All figures where this applies: please indicate the location of the binding sites from the structural data" has been ignored by the authors without any comment.I still strongly suggest amending the figures accordingly as this would make them more accessible.

We thank the reviewer for the suggestion. As requested, we now have added blue arrows to Figure 2 to denote the locations of the ion-binding sites resolved in the cryo-EM density maps. For the time traces shown in Figure 2—figure supplement 2 and Figure 3, the y-axis (ion position along pore axis) is too compressed to allow readers to distinguish the individual binding sites. We hope the reviewer understands why we cannot include the arrows in these panels.

This comment of mine has maybe been overlooked by the authors: What is the meaning of the numbers in the bracket in Figure 3A? I presume the number of water molecules?

The reviewer is correct: the number in the brackets correspond to the number of water molecules in the primary shell of the ion in the snapshot. The figure legend has been revised accordingly.

Reviewer #2 (Recommendations for the authors):All points have been adequately addressed. I recommend the paper for publication.

We thank the reviewer for their helpful comments during the review process that have greatly improved this manuscript.

Reviewer #3 (Recommendations for the authors):The authors have addressed most of the comments appropriately.

We thank the reviewer for their helpful comments during the review process and have addressed their remaining comment in the revised manuscript.

In their response concerning the choice of forcefield, they write: "It could be convincingly argued that a mechanism that relies on dehydration energetics, rather than specific ion-protein interactions, will be reasonably described by this forcefield, as it is in fact parametrized, with excellent accuracy, to reproduce the relative free energies of hydration (and thus dehydration) of the alkali cations." That is the kind of justification that I was hoping for. Please make that convincing argument in the text, including references as needed.

We thank the reviewer for the suggestion and have included a short discussion of the forcefield used in our work in the revised manuscript.